# Learning how network structure shapes decision-making for bio-inspired computing

Michael Schirner [1,2,3,4,5] ✉, Gustavo Deco [6,7,8,9] & Petra Ritter [1,2,3,4,5] ✉

To better understand how network structure shapes intelligent behavior, we developed a learning algorithm that we used to build personalized brain network models for 650 Human Connectome Project participants. We found that participants with higher intelligence scores took more time to solve difficult problems, and that slower solvers had higher average functional connectivity. With simulations we identified a mechanistic link between functional connectivity, intelligence, processing speed and brain synchrony for trading accuracy with speed in dependence of excitation-inhibition balance. Reduced synchrony led decision-making circuits to quickly jump to conclusions, while higher synchrony allowed for better integration of evidence and more robust working memory. Strict tests were applied to ensure reproducibility and generality of the obtained results. Here, we identify links between brain structure and function that enable to learn connectome topology from non-invasive recordings and map it to inter-individual differences in behavior, suggesting broad utility for research and clinical applications.

Do intelligent people think faster? Strong correlations between reaction times and intellectual performance support this idea, providing a cornerstone for intelligence research for over one century[1–6]. Here, we show an important exception in empirical data and provide an explanation based on brain simulation (Supplementary Movie 1). Participants with higher intelligence were only faster when the test was simple. Conversely, in hard tests that required problem solving over several seconds or minutes without time limit, participants with higher intelligence used more, not less time to arrive at correct solutions. We reproduced this link between reaction time and performance in personalized multi-scale brain network models[7,8] (BNMs) that couple each participant's structural white-matter connectivity (SC) with a generic neural circuit for decision-making (DM) and working memory (WM). Simulation results indicate that decision-making speed is traded with accuracy, resembling influential theories

from the fields of economy and psychology on fast and slow thinking[9].

Intelligence is here defined as the performance in psychometric tests in cognitive domains like verbal comprehension, perceptual reasoning or working memory. A consistent finding is that individuals who perform well in one domain tend to perform well in the others, which led to the derivation of a general factor of intelligence called g-factor[10]. While the g-factor also targets learned skills like verbal fluency, the term fluid intelligence (FI) refers to abilities related to solving new problems independently of acquired knowledge[11]. Reaction time (RT) as a measure of cognitive processing speed provides strong evidence in support of the idea that people are more intelligent because they have faster brains[2]. A meta-analysis over 172 studies and 53,542 participants reported strong negative correlations between general intelligence and diverse measures of RT[6]. RT and intelligence are also linked

[1]Berlin Institute of Health (BIH) at Charité – Universitätsmedizin Berlin, Charitéplatz 1, 10117 Berlin, Germany. [2]Department of Neurology with Experimental Neurology, Charité, Universitätsmedizin Berlin, Corporate member of Freie Universität Berlin and Humboldt Universität zu Berlin, Charitéplatz 1, 10117 Berlin, Germany. [3]Bernstein Focus State Dependencies of Learning and Bernstein Center for Computational Neuroscience, Berlin, Germany. [4]Einstein Center for Neuroscience Berlin, Charitéplatz 1, 10117 Berlin, Germany. [5]Einstein Center Digital Future, Wilhelmstraße 67, 10117 Berlin, Germany. [6]Department of Information and Communication Technologies, Center for Brain and Cognition, Computational Neuroscience Group, University of Pompeu Fabra, Barcelona, Spain. [7]Catalan Institution for Research and Advanced Studies, Barcelona, Spain. [8]Department of Neuropsychology, Max Planck Institute for Human Cognitive and Brain Sciences, Leipzig, Germany. [9]School of Psychological Sciences, Turner Institute for Brain and Mental Health, Monash University, Clayton, Melbourne, VIC, Australia. ✉e-mail: michael.schirner@bih-charite.de; petra.ritter@bih-charite.de

over the lifespan: RT increases with age and is strongly correlated with decline in other domains[5,12]. Intriguingly, RT is a more powerful predictor of death than well-known risk factors like hypertension, obesity, or resting heart rate: RT is the second most important predictor of death after smoking[13] and explains two-thirds of the relationship between general intelligence and death[14]. After adjusting for smoking, education, and social class, RT was an even stronger predictor of death than intelligence. However, these results do not imply that PS is the causal factor underlying intelligence: an important counterargument is that training and improving PS does not transfer to untrained measures[15].

We found that participants with higher intelligence were only quicker when responding to simple questions, while they took more time to solve hard questions. This became apparent in the Penn Matrix Reasoning Test (PMAT), which consists of a series of increasingly difficult pattern matching tasks for quantifying FI[11]. While PS tests are typically so simple that people would not make any errors if given enough time, FI tests like PMAT can be unsolvable even without time limit. PMAT requires to infer hidden rules that govern the figure, which involves a recursive decomposition of complex problems into easier subproblems, forming a hierarchy of DM processes[11]. To solve the problem, it is required to make decisions about tentative solution paths while storing previous progress in WM. Sub-problems higher up in the hierarchy need to be held longer in WM as evidence from lower in the hierarchy needs to be integrated later in time[11]. Therefore, taking decisions on higher-level problems must be held out until evidence from sub-problems was integrated to not prematurely jump to a conclusion. This form of cognition can be contrasted with the flexibility required by PS tests where it is actually advantageous if decisions do not rely on extensive accumulation of evidence and memories can be flexibly overwritten.

Here, by closely fitting brain models to each subject's functional connectivity (FC), we identify a fast mode of cognition for rapid decision-making and flexible working memory and contrast it with a slow mode of cognition that supports prolonged integration of information and more stable working memory. Importantly, by identifying a smooth and monotonous relationship between structural and functional neural network architecture it was possible to devise a

network fitting algorithm that allows to simultaneously and precisely control the state of synchronization between every pair of network nodes, allowing to tune each connection from full antisynchronization to full synchronization, enabling a close reproduction of whole-brain subject-specific FC. In the following, we first provide behavioral findings that link intelligence test results with processing speed and FC (Fig. 1 and Table 1). Then we demonstrate a computational framework for closely fitting BNMs to personal FC (Figs. 2 and 3), and subsequently explain the empirical data based on the in silico identified biological candidate mechanisms (Figs. 4–6 and Supplementary Figures). For the fitting we created a parameter learning algorithm that makes use of our observation that FC and synchronization between two simulated brain areas can be smoothly and monotonically tuned via their long-range excitation-inhibition balance (E/I-ratio). We then show that the internal dynamics of the fitted models correlated with the empirical cognitive performance of the subjects (Fig. 4a, b). In addition, E/I-balance modulated the amplitude and synchrony of large-scale synaptic currents in a way that modulated DM winner-take-all races and WM persistent activity in accordance with the empirical observations (Figs. 5 and 6 and Supplementary Fig. 4). Phase space analysis of the resulting model dynamics allowed to frame the trade-off between speed and accuracy in terms of generic dynamical systems behavior in dependence of the E/I-balance of long-range brain network topology, which may jointly explain individual variability in FC, intelligence, and processing speed (Supplementary Figs. 5 and 6 and Supplementary Movie 1).

## Results
### Higher intelligence: taking complex decisions slowly
We analyzed correlations between *g*-factor, FI (PMAT24_A_CR), RT for correct responses in the FI test (PMAT24_A_RTCR), and processing speed for 1176 participants of the Human Connectome Project (HCP) Young Adult study (Table 1)[16]. FI was measured by the number of correct responses in PMAT (PMAT24_A_CR)[11,17]. Processing speed was measured by the NIH Toolbox tests Dimensional Change Card Sort[18] and Pattern Completion Processing Speed[19] (CardSort_Unadj and ProcSpeed_Unadj). For findability we use the same abbreviations for the cognitive tests as used in the HCP (Table 2).

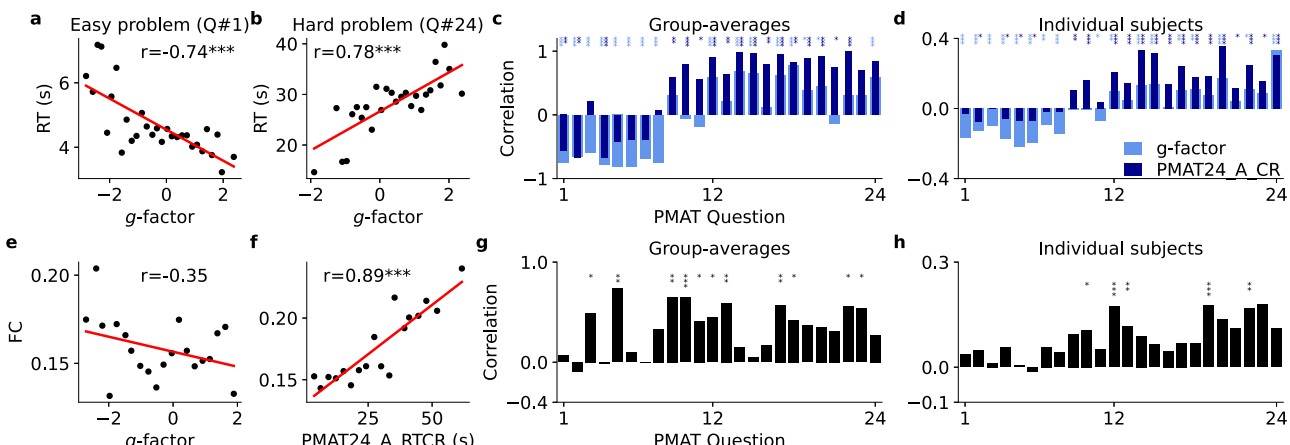

**Fig. 1 | Correlations between intelligence, RTs and FC. a, b** Group-average *g*-factor (30 groups, based on g-factor, *N* = 650 subjects) versus RT for correct responses in PMAT questions #1 (very easy, *p* = 4.0 × 10⁻⁶) and #24 (very hard, *p* = 3.0 × 10⁻⁶). **c, d** Group-average and subject-level correlations between *g*/PMAT24_A_CR and the RT for correct responses in each individual PMAT question. Subjects with higher *g*/PMAT24_A_CR were quicker to correctly answer easy questions, but they took more time to correctly answer hard questions (questions sorted according to increasing difficulty; sign of correlation flips at question #9). **e** Group-average *g*-factor versus mean FC (20 groups, based on *g*-factor,

*N* = 650 subjects, *p* = 0.13). **f** Group-average PMAT24_A_RTCR versus mean FC (20 groups, based on PMAT24_A_RTCR, *N* = 650 subjects, *p* = 6.9 × 10⁻⁷).
**g, h** Group-average (20 groups, based on PMAT24_A_RTCR) and subject-level correlations between mean FC and RT for correct responses in each PMAT question. Subjects that took more time to correctly answer test questions had a higher FC, independent of whether the question was easy or hard. *P* values of two-sided Pearson's correlation test: \**p* < 0.05, \*\**p* < 0.01, \*\*\**p* < 0.001; including only *p* values that remained significant after controlling for multiple comparisons using the Benjamini–Hochberg procedure with a False Discovery Rate of 0.1.

**Table 1 | Correlation coefficients between intelligence, RT, and PS on an individual-subject level (N = 1176)**

| Pearson correlation (N = 1176) | g | PMAT24_A_CR | PMAT24_A_RTCR | CardSort_Unadj |
|---|---|---|---|---|
| PMAT24_A_CR | 0.79* | | | |
| PMAT24_A_RTCR | 0.5* | 0.72* | | |
| CardSort_Unadj | 0.43* | 0.23* | 0.03 | |
| ProcSpeed_Unadj | 0.25* | 0.16* | −0.04 | 0.42* |

Abbreviations of cognitive tests are introduced in Table 2. Note that PMAT24_A_RTCR measures RTs (larger values indicate longer times), while CardSort_Unadj and ProcSpeed_Unadj are PS tests that measure the inverse of time (larger values indicate shorter times), hence signs of the correlation coefficients are reversed. P values of two-sided Pearson's correlation test: *p < 0.001.

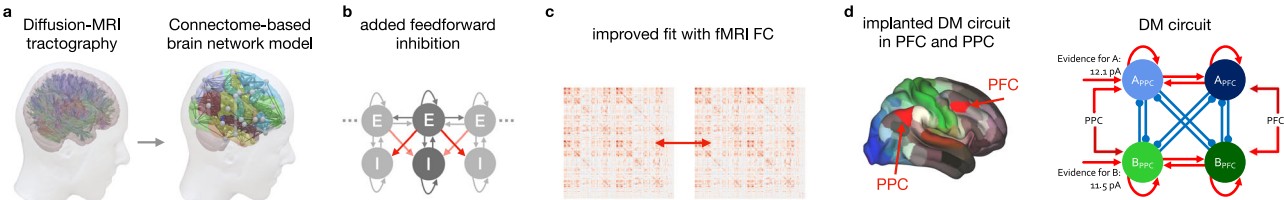

**Fig. 2 | Modeling outline. a** 379-nodes large-scale BNMs were constructed from person-specific white matter connectomes estimated with dwMRI tractography. In addition, a simplified network with only two nodes (but identical node dynamics) was used to create E/I-ratio tuning curves (Fig. 4). **b** In previous BNM studies long-range white matter coupling from excitatory to inhibitory populations was often absent. Adding these connections allowed to tune the relative strength of long-range excitatory-to-excitatory versus long-range excitatory-to-inhibitory connections, enabling to precisely tune the E/I-ratio of synaptic inputs between each pair of BNM nodes. Importantly, setting the E/I-ratio allowed to monotonically and smoothly control the FC between all nodes (Fig. 3a). Underlying predicted fMRI time series, the E/I-ratio allowed to smoothly tune synchronization and amplitude of synaptic currents (Fig. 4). **c** By systematically tuning E/I-ratios, the fit between simulated and empirical FC can be increased until full similarity (Fig. 3b, c). **d** Upon fitting each participant's BNM with their empirical FC, each BNM was coupled with a smaller scale frontoparietal circuit for simulating DM and WM. Subpopulations in prefrontal cortex (PFC) and posterior parietal cortex (PPC) are mutually and recurrently coupled to encode two decision options A and B. For example, evidence for option A recurrently excited the populations $A_{PPC}$ and $A_{PFC}$ (red connections) while it led to an inhibition of the populations $B_{PPC}$ and $B_{PFC}$ (blue connections). Importantly, instead of independent noise, we used the activity of the PFC and PPC regions of the 379-nodes large-scale network to drive the DM circuit, which allowed to analyze how local decision-making and working memory performance can be modulated by large-scale brain network topology. Panel **a** is adapted from ref. 77. and used under a CC BY 4.0 license (https://creativecommons.org/licenses/by/4.0/).

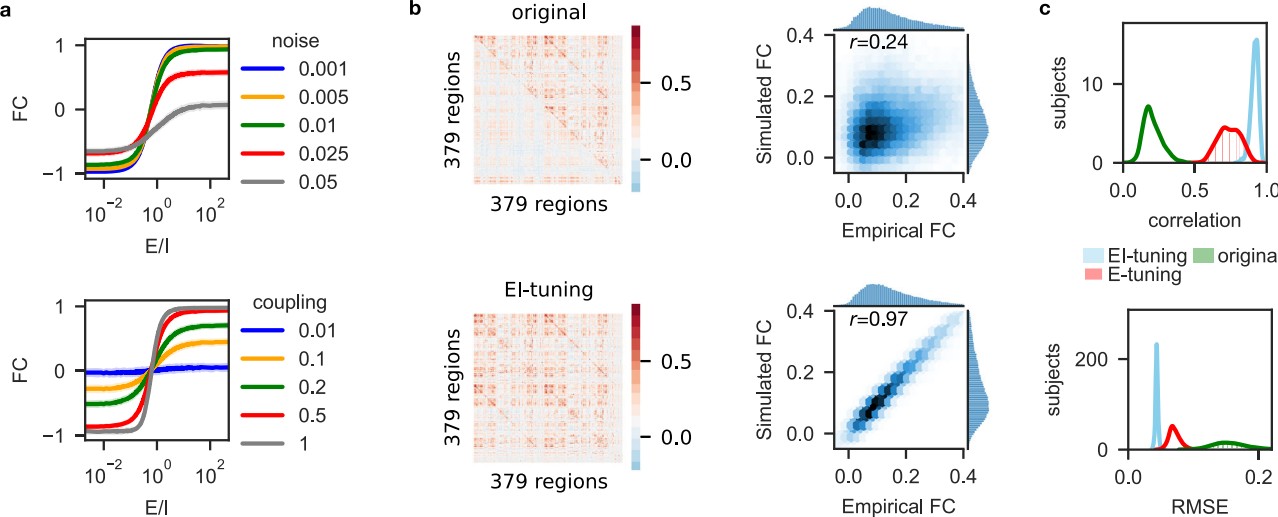

**Fig. 3 | Identification of a smooth, monotonic relationship between E/I-ratio and FC to fit brain network models. a** Tuning curves for a reduced model with only two nodes, but otherwise identical to the 379-nodes BNM. FC (that is, correlation) between the two nodes increased smoothly and monotonically as a function of their E/I-ratio $\frac{w_{1,2}^{LRE}}{w_{1,2}^{FFI}}$. The relationship between E/I-ratio and FC persisted when the strength of noise $\sigma$ (upper panel; Eqs. 5 and 6) and the strength of structural coupling $C_{ij}$ (lower panel; Eqs. 1 and 2) were modulated for test purposes (both are fixed parameters during the fitting of the full 379-nodes model). **b** Fitting results for the full 379-nodes model for one exemplary FC. Empirical (upper triangular portion of the matrix) versus simulated (lower triangular portion of the matrix) FC and joint distributions without E/I-tuning (upper panel) and with E/I-tuning (lower panel). **c** Pearson correlations and root-mean-square errors between all N = 650 empirical and simulated FCs for three different model variants: EI-tuning (the tuning algorithm applied on both $w_{ij}^{LRE}$ and $w_{ij}^{FFI}$), E-tuning (the tuning algorithm applied only on $w_{ij}^{LRE}$), original (tuning of a scalar global coupling scaling factor to rescale $C_{ij}$).

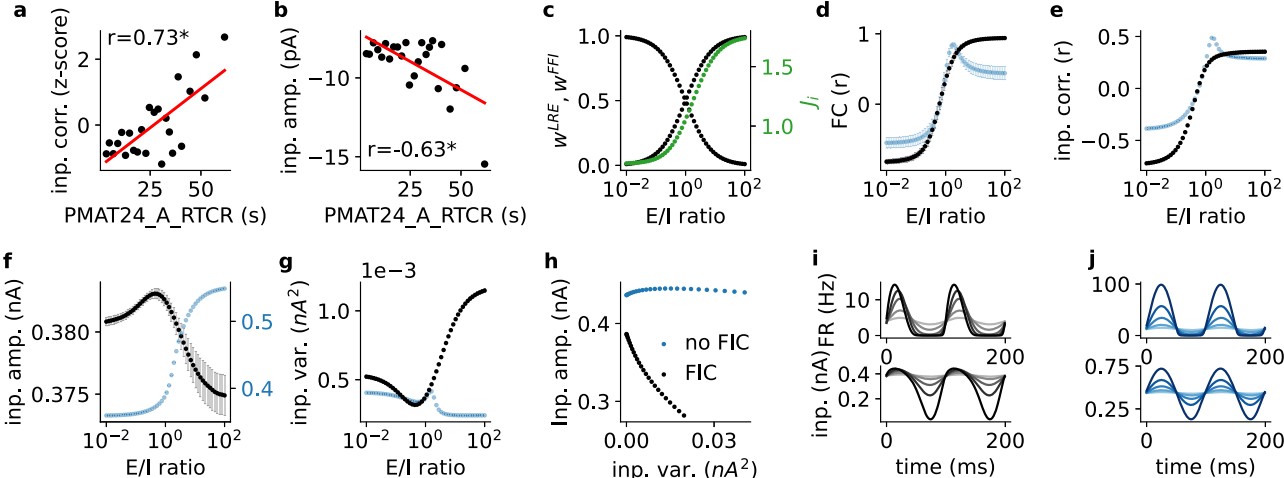

**Fig. 4 | Model dynamics correlate with empirical cognitive performance. FC, synchrony, amplitude and variance of neural population activity depend on E/I-ratios. a** PMAT24_A_RTCR versus strength of correlation of input currents in the full 379-nodes large-scale BNMs. The models of slower subjects had a higher synchrony between the time series of synaptic currents $I_i^E$ ($p = 9.8 \times 10^{-5}$). **b** PMAT24_A_RTCR versus input amplitude. The models of slower subjects had a lower average synaptic current amplitude $I_i^E$ ($p = 8.6 \times 10^{-4}$)). **c** E/I-ratio versus parameter settings in the simplified two-node large-scale model. The E/I-ratio of a connection is defined by the quotient of long-range excitation $w_{ij}^{LRE}$ (black) and feedforward inhibition $w_{ij}^{FFI}$ (black). $J_i$ values (green) were obtained by FIC. **d** E/I-ratio versus FC for active (black) and inactive FIC (blue). A monotonic relationship between E/I-ratio and FC only emerged when FIC was active. **e** E/I-ratio versus correlation of input currents. **f** E/I-ratio versus input amplitude. With FIC input amplitudes peaked at relatively low E/I-ratios and then continued to monotonically decrease for increasing E/I-ratios **g** E/I-ratio versus input variance showed an

inverse pattern compared to **f. h** Amplitude versus variance of inputs. FIC coupled the variance of synaptic inputs with the amplitude of synaptic inputs: the higher the variance (resulting from stronger coupling), the lower the amplitude. **i, j** Firing rate (Eq. 3) and input current (Eq. 1) time series after injecting 10-Hz sinusoidal waves with increasing variance for active (black) and inactive FIC (blue). FIC compensated higher input variances (which were modulated by the fitting algorithm via the multiplicative coupling parameters $w_{LRE}$ and $w_{FFI}$) with a lower mean (**h**). This was necessary as the upper half-wave of the input continued to grow in amplitude for increasing E/I-ratios, while the lower half-wave was bounded by 0 Hz firing (gray to black lines), which required FIC to increase $J_i$ to arrive at the same target average firing rate of 4 Hz. Data in panels c-h are presented as mean values +/- SD derived from N = 100 simulations with different random number generator seeds. Obtained $p$ values of two-sided Pearson's correlation test: *$p < 0.001$; including only $p$ values that remained significant after controlling for multiple comparisons using the Benjamini-Hochberg procedure with a False Discovery Rate of 0.1.

Reproducing established results[6], individuals with higher $g$ and FI (PMAT24_A_CR) were faster in the simple processing speed tests. However, they needed more, not less time (PMAT24_A_RTCR) to form correct decisions in the harder FI test (PMAT24_A_CR, Table 1). This observation is remarkable as it challenges the notion that higher intelligence is the result of a faster brain.

The observation may however have a trivial explanation: PMAT questions are arranged in order of increasing difficulty and the test is discontinued if the participant makes five incorrect responses in a row. People with higher intelligence could have a higher RT simply because they advanced until the more difficult questions. To exclude this explanation, we correlated intelligence with the RTs for each individual PMAT question, which shows the impact of problem difficulty on RT: for the first eight questions participants with higher $g$ and PMAT24_A_CR were faster to give correct answers, but slower for the remaining sixteen questions (Fig. 1a–d).

**Slow solvers have higher resting-state functional connectivity**
Next, we compared cognitive performance with mean FC (average correlation between all region-wise fMRI time series) in a subset of N = 650 participants with complete data and where no quality control issues were identified by the HCP consortium (see Methods). We have selected mean FC for the subsequent analyses as it is a compact representation of whole-brain FC and related to E/I-balance per our analysis (Figs. 3 and 4). Mean FC had no significant correlation with g on single-subject level ($r = 0.02$, $p = 0.69$) and group level (Fig. 1e and Supplementary Fig. 1a). On the single-subject level there was a significant correlation between mean FC and PMAT24_A_RTCR ($r = 0.13$, $p = 0.0012$). Multiple regression to compute the coefficient of multiple correlation between all reported behavioral variables ($g$, PMAT24_A_CR, PMAT24_A_RTCR, ProcSpeed, CardSort) and mean FC

yielded $r = 0.16$ ($p < 0.001$), which was only slightly higher than the univariate correlation between mean FC and PMAT24_A_RTCR.

Importantly, independent of the complexity of the question there were strong positive correlations between mean FC and the times to correctly answer each individual PMAT question (Fig. 1g, h): slower participants tended to have higher mean FC, regardless of whether the question was easy or hard, indicating that FC (or properties of the brain network underlying FC) could be related to the modulation of processing speed, which we studied with computational models below.

**Excitation-inhibition balance controls functional connectivity**
Which neurophysiological processes underly the observed correlations between intelligence, RT, and FC? To study neuronal processing in silico we created BNMs for the 650 subjects using a tuning algorithm that fits each participant's simulated FC with their empirical FC (Figs. 2 and 3). The BNMs use coupled neural mass models to simulate the electric, synaptic, firing, and hemodynamic (fMRI) activity of a 379-nodes whole-brain network. Each node consists of one excitatory and one inhibitory population that mutually and recurrently interact. To simulate long-range white matter coupling, the neural masses were connected by each participant's SC, which were estimated by dwMRI tractography. Importantly, we added feedforward inhibition to increase biological realism[20–30]: while in previous BNM studies there was typically only long-range coupling between excitatory populations, here, excitatory masses additionally targeted inhibitory populations (Fig. 2b and "Methods"). In addition, the strength of local inhibitory feedback from the inhibitory to the excitatory population of each node was controlled by inhibitory synaptic plasticity[31], which was set to tune each excitatory population's long-term average firing rate to 4 Hz in a process called Feedback Inhibition Control (FIC)[32]. By tuning the ratio of long-range excitation (LRE; strength of long-range

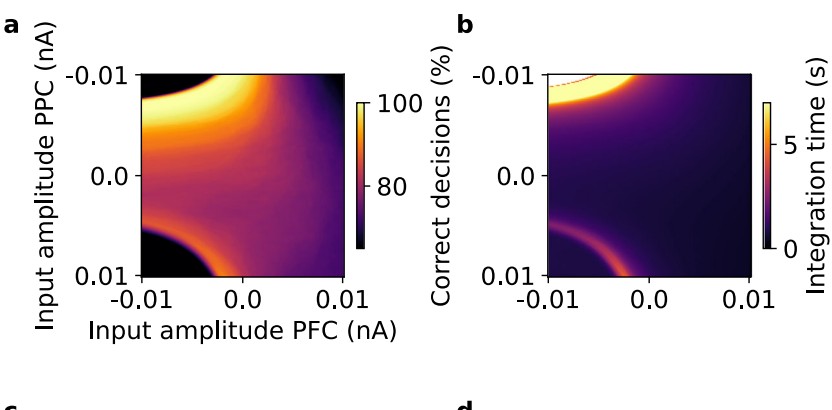

**Fig. 5 | DM performance depends on amplitude and synchrony of input currents to the isolated frontoparietal DM circuit. Decreased amplitude of PFC and PPC noise and increased synchrony of PPC noise led to more correct decisions and longer integration time in the DM circuit. a** Percent correct decisions for varying the mean amplitudes of the input noise time series to the PFC and PPC modules of the DM circuit $I_{noise,i}^{PFC}$ and $I_{noise,i}^{PPC}$. **b** Evidence integration times for varying mean amplitudes of the input noise time series $I_{noise,i}^{PFC}$ and $I_{noise,i}^{PPC}$. **c** Percent correct decisions for varying correlation coefficients between input noise time series $I_{noise,i}^{PFC}$ and $I_{noise,i}^{PPC}$. **d** Evidence integration times for varying correlation coefficients between input time series $I_{noise,i}^{PFC}$ and $I_{noise,i}^{PPC}$.

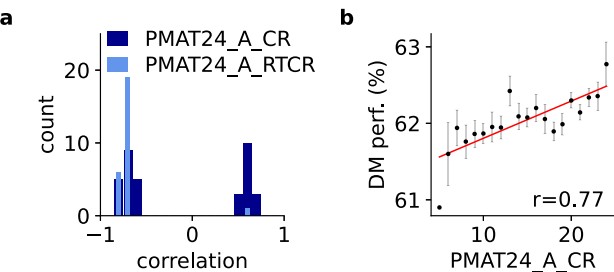
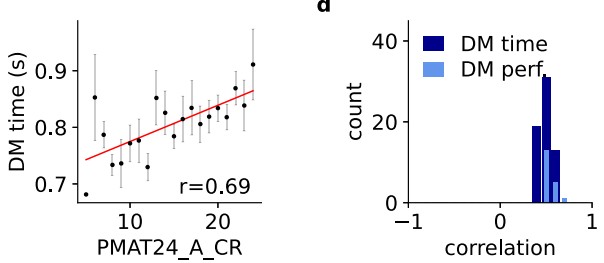

**Fig. 6 | Multiscale modeling: coupling PFC and PPC nodes of the person-specific BNMs with the corresponding modules of the generic DM circuit. The models of subjects with higher PMAT24_A_CR (fluid intelligence) made fewer mistakes, but were slower, echoing the empirically observed trade-off. a** Distribution of significant correlations between mean input of all BNM nodes and PMAT24_A_CR ($p < 0.05$ for 35 of 379 nodes), respectively PMAT24_A_RTCR ($p < 0.05$ for 26 of 379 nodes) over all $N = 650$ models. **b, c** Group-average PMAT24_A_CR versus DM performance ($r = 0.77$, $p = 7.2 \times 10^{-5}$), respectively DM time ($r = 0.69$, $p = 7.2 \times 10^{-4}$), for an exemplary combination of PFC and PPC nodes. Data are presented as mean values +/− SD over all $N = 650$ models each simulated 100 times with different random number generator seeds. **d** Distribution of significant correlations between group-average PMAT24_A_CR and DM time ($p < 0.05$ for 57 of 90 possible combinations), respectively DM performance ($p < 0.05$ for 19 of 90 possible combinations) over all $N = 650$ models. Including only correlations that remained significant after controlling for multiple comparisons using the Benjamini-Hochberg procedure with a False Discovery Rate of 0.1.

## Table 2 | Abbreviations of the used cognitive tests

| Test | Description | References |
|---|---|---|
| *g, g*-factor | General factor of intelligence derived using a bi-factor model of intelligence[82] | 84 |
| PMAT24_A_CR | Number of correct responses in Penn Progressive Matrices. Nonverbal estimate of fluid intelligence using an abbreviated version of Raven's standard progressive matrices test | 17 |
| PMAT24_A_RTCR | Median response time (ms) for correct responses. Larger values for higher RT | 17 |
| CardSort_Unadj | Score in NIH Toolbox Dimensional Change Card Sort test, a measure of cognitive flexibility. Smaller values for higher RT | 18,85 |
| ProcSpeed_Unadj | Score in Pattern Completion Processing Speed test. Smaller values for higher RT | 19 |

excitatory-to-excitatory coupling $w_{ij}^{LRE}$, Eq. 1) to feedforward inhibition (FFI; strength of long-range excitatory-to-inhibitory coupling $w_{ij}^{FFI}$, Eq. 2) between each pair of brain regions it was possible to precisely control the synchrony, respectively functional connectivity, of the entire brain network (Fig. 3 and Supplementary Movie 1). Although many parameters are simultaneously tuned, which may raise concerns about overfitting, we show below that the fitting procedure robustly predicts the same model dynamics over different re-initializations and that the fitted models produce generalizable mechanistic insights and meaningfully comparable predictions over the subject cohort. While in previous models the values of $w_{ij}^{LRE}$ and $w_{ij}^{FFI}$ were implicitly set to the same scalar constant for every pair of brain regions, with this approach E/I ratios can be justified in a principled and data-driven way, in agreement with the direct relationship that we identified between E/I ratios and FC: increasing E/I-ratios led to increasingly positive FC up to full synchronization; vice versa, decreasing E/I-ratios decreased the correlation between the simulated fMRI time series until full anti-synchronization (Fig. 3a). By simultaneously tuning the E/I-ratios of every connection to minimize the error between empirical and simulated FC, it was possible to considerably improve FC fits to a point where the simulated FCs of all 650 individual BNMs became almost indistinguishable from their empirical counterparts, explicitly reproducing even intricate and subtle patterns (Fig. 3b). In comparison to the original model (Fig. 3c, green curves), where E/I-ratios were left untuned at their default settings ($w_{ij}^{LRE} = 1$ and $w_{ij}^{FFI} = 0, \forall i, j \in \{1, \ldots, N\}$) from Deco et al.[32], and compared to a variant where only $w_{ij}^{LRE}$ values were tuned (Fig. 3c, red curves), tuning both $w_{ij}^{LRE}$ and $w_{ij}^{FFI}$ at the same time allows to smoothly set the state of synchronization between each pair of brain regions (Figs. 3b and 4d, e), which can be used to considerably reduce the root-mean-square error between simulated and empirical FC (Fig. 3c, blue curves). It is important to point out that E/I-ratio here refers only to the ratio of the long-range coupling strength parameters $\frac{w_{ij}^{LRE}}{w_{ij}^{FFI}}$ without considering the effect of local inhibitory connectivity $J_i^i$. Due to FIC the E/I-ratio of the total sums of long-range and local currents that arrive at excitatory populations ($\frac{W_E I_0 + w_+ J_{NMDA} S_i^E + J_{NMDA} \sum_j w_{ij}^{LRE} C_{ij} S_j^E}{J_i^i S_i^I}$, Eq. 1) is always in a balanced state, which ensures an average firing rate of 4 Hz of the excitatory population even in the case that long-range connections are unbalanced.

Summarizing, the long-range E/I-ratios between network nodes control the direction (positive versus negative) and strength of their synchronization and FC; tuning these E/I-ratios enables simulation of person-specific empirical FCs with average correlations $r > 0.97$.

### Simulated brain activity correlates with cognitive performance
To identify processes relevant for intelligence, we correlated the dynamics of each subject's BNM with their PMAT24_A_RTCR. On the single-subject level we found only a low negative correlation between PMAT24_A_RTCR and the mean amplitude of synaptic currents ($r = −0.11$, $p = 0.0068$) and a low positive correlation with the mean correlation between synaptic currents ($r = 0.13$, $p < 0.001$). For the two processing speed measures CardSort_Unadj and ProcSpeed_Unadj no significant correlations were obtained on the single-subject level.

On the group-average level correlations with PMAT24_A_RTCR were however large showing that the models of slower subjects had on average a lower amplitude of synaptic currents, but a higher synchrony between synaptic currents (Fig. 4a, b and Supplementary Fig. 2). Importantly, synaptic currents had an almost linear relationship with FC on an individual-subject level (Supplementary Fig. 3), indicating that E/I-ratios also control amplitude and synchrony of synaptic currents, which possibly points towards brain network mechanisms for explaining the observed differences in cognition. To better isolate the involved mechanisms, we again studied a reduced version of the 379-nodes BNM with only two-nodes.

### How E/I-ratios control FC
To study how E/I-ratios modulate FC in isolation we tuned E/I-ratios from 0.01 to 100 in the two-node model. The two-node model is a simplified version of the 379-node large-scale brain model to study the effect of large-scale E/I-balance with a simpler network structure (Fig. 4c–j). The two-node model (Eqs. 1–6) differed from the functional frontoparietal decision-making circuit[33] (DM circuit, Eqs. 7–10) further introduced below. The two-node model simulated mutual and recurrent interaction between one excitatory and one inhibitory population as in the 379-nodes large-scale model, but with a simpler network of only two nodes to produce tuning curves (Fig. 4c–h). In contrast, the DM circuit is an existing frontoparietal circuit model to simulate winner-take-all competition resulting from cross-inhibition of two excitatory populations via one inhibitory population, which we studied in isolation (Fig. 5), and after coupling with the 379-nodes large-scale model to form the multiscale model (Fig. 6). Dynamics of the two-node model were identical to the full 379-regions model but with only two nodes $i, j$ that had a mutual coupling strength of $C_{ij} = C_{ji} = 1$. To increase E/I-ratios we increased $w^{LRE}$ and decreased $w^{FFI}$ under the constraint $w^{LRE} + w^{FFI} = 1$ to keep the total sum of inputs constant (Fig. 4c). As before, FIC was used to tune average firing-rates of the excitatory populations to a biologically plausible rate of 4 Hz[32]. As before (Fig. 3a), increasing the E/I-ratio increased FC from a strong negative to a strong positive correlation (Fig. 4d). Underlying the simulated fMRI, also the correlation between simulated synaptic inputs increased monotonically from negative to positive (Fig. 4e). This monotonic relationship enabled fitting the models to empirical FC using a simple learning rule that increased or decreased E/I-ratios based on the strength of FC of each connection (Methods). Interestingly, the monotonic relationship only emerged when FIC was active (Fig. 4d, e, black curves). When FIC was disabled (Fig. 4d, e, blue curves), a complex nonlinear relationship between E/I-ratios and FC appeared, which would prevent the fitting with empirical FC. That is, without FIC, increasing the E/I-ratio could either increase or decrease the FC, and vice versa, while with FIC FC can be smoothly increased by increasing the E/I-ratio and vice versa. These observations underline the importance of FIC: only when FIC was active synaptic correlations increased and synaptic amplitude decreased for increased E/I-ratios, respectively FC (Fig. 4d–f). Therefore, only with FIC a concordant effect of amplitude and correlation on decision times and decision accuracy was obtained that is in line with empirical data. Supplementary section How E/I-ratios control synchrony and amplitude of synaptic currents describes the involved mechanisms in more detail.

### E/I-ratios switch between fast and accurate DM
To better understand how E/I-ratios modulate DM and WM we used an existing[33] frontoparietal circuit model for winner-take-all DM and persistent activity WM called DM circuit in the following (see Supplementary section Studying DM and WM with a frontoparietal circuit model). In the DM circuit NMDAergic and GABAergic synaptic dynamics of prefrontal cortex (PFC) and posterior parietal cortex (PPC) decision populations are explicitly modeled, while uncorrelated and independent noise from an Ornstein-Uhlenbeck process is used to simulate AMPA synapses[33]. However, a more realistic assumption is that synaptic inputs are not uncorrelated, but that populations receive correlated inputs from shared presynaptic groups[34–40]. Furthermore, inputs might not necessarily be fully balanced and centered at zero. Rather, our BNM simulations suggest that input amplitudes and correlations vary heterogeneously across brain areas and subjects and are strongly related to FC (Supplementary Fig. 3). Consequently, we systematically varied amplitude and correlation of AMPA noise inputs and found that they switch the DM circuit between fast-but-faulty and precise-but-slow modes of DM (Fig. 5). Decreasing the mean amplitude of inputs increased decision accuracy as well as integration time (Fig. 5a, b). Similarly, increasing the correlation of input noise to the

two PPC populations also led to increased performance and integration time (Fig. 5c, d). Integration times followed an inverted U-shape and were at their maxima for intermediate levels of noise correlation ($r \sim 0.5$, Fig. 5d). In contrast, input correlation to the two PFC populations had no relevant effects (Fig. 5c, d). These results indicate that DM performance depends on synaptic inputs in line with our empirical data: participants with higher FC (corresponding to lower amplitudes, but higher input correlations in the model) were slower (Fig. 1 g, h) but made fewer errors (Fig. 1c, d). They also corroborate the identified link between empirical PMAT24_A_RTCR and synaptic inputs in the BNM simulations, where higher input synchrony and lower input amplitudes correlated with longer PMAT24_A_RTCR (Fig. 4a, b and Supplementary Fig. 2). The underlying dynamic mechanisms are described in supplementary sections How input amplitude modulates DM performance and How input correlation modulates DM performance (see also Supplementary Figs. 4 and 5 and Supplementary Movie 1).

### E/I-ratios switch between stable and flexible WM

We also tested the effect of input amplitude on WM in the DM circuit and created bifurcation diagrams that visualize dynamical regimes of the system as a function of net recurrent synaptic currents $J_S$ (recurrent excitation minus cross-inhibition) and stimulus strength $I_{app}$ (Supplementary Fig. 6). Memories were induced by a brief stimulus to one of the PPC populations, which created persistent activity in the memory-encoding population. At $t = 1.5$ s after the target stimulus a distracting stimulus was applied to the other population, to test the robustness of the memory-encoding persistent activity. The WM state was robust if the memory-encoding population maintained its persistent high firing activity and it was fragile if the persistent firing was disrupted. Varying $J_S$ and $I_{app}$ parameters gave rise to three dynamical regimes in the bifurcation diagram: robust WM, disrupted WM, or no induction of WM at all (Supplementary Fig. 6). We found that the thresholds for WM induction and robustness shifted in dependence of input amplitude. Decreasing the input amplitude increased the thresholds for WM induction and disruption, which in turn requires larger stimuli to induce or overwrite WM content (Supplementary Fig. 6). A decreased input amplitude therefore makes WM less flexible, which is again in line with our empirical observations: slower subjects had a higher FC (Fig. 1f–h and Supplementary Fig. 1), which was related to decreased input amplitude via BNM simulations (Fig. 4b and Supplementary Figs. 2a and 3a) and two-node model simulations (Fig. 4d, f). Vice versa, higher input was related to lower thresholds for the induction and overwriting of working memories, which made WM more flexible to support simple but time-sensitive tasks.

### Coupling the DM circuit with the large-scale BNMs

To predict DM performance of each individual, we coupled the DM circuit with each of the 650 BNMs with the effect that the PFC and PPC modules of the DM circuit were driven by large-scale PFC and PPC inputs instead of the independent noise that was used in the isolated circuit (replacing Eq. 7a from the original DM circuit model[33] by Eq. 7b). Correlations between PMAT24_A_RTCR, respectively PMAT24_A_CR, and the input amplitudes of the 379 BNM regions indicate that the amplitudes encode information about individual cognitive performance (Fig. 6a). For coupling we identified 10 PFC and 9 PPC atlas regions[41] that were activated during n-back task performance, which combines aspects of WM and DM (PFC: a9-46v, 9-46d, p9-46v, 8 C, i6-8, s6-8, 8Av, SFL, and 8BM. PPC: AIP, LIPd, IP1, IP2, 7PL, 7Pm, 7Am, POS2, PFm, and PGs). Simulation results predicted empirical performance for several of the 90 possible PPC-PFC combinations (Fig. 6b–d). Multiscale models of participants with higher FI (PMAT24_A_CR) also had a higher DM accuracy and needed more time to take the decisions, reproducing the empirical data.

### Model validation

To test the robustness of the fitting procedure we ran it 1000 times with random initial conditions and noise generator seeds using the average SC and FC from all subjects. The minimum correlation between all 1000 simulated FCs was $r = 0.9946$ and their average correlation with the empirical FC was $r = 0.9973$, which shows that the procedure consistently led to a high fit. Next, we simulated one hour of fMRI with the 1000 fitted models, this time using the same noise. The average correlation between all resulting fMRI time series over all 379 brain regions was $r = 0.9962$, showing that the fitting led to consistent fMRI predictions although there existed a variance in the obtained model parameters (average coefficients of variation $CV_{LRE} = 0.5$ and $CV_{FFI} = 0.72$). Although the repeated fitting runs did not converge to a unique parameter set the simulated time series were nevertheless robustly reproduced as a general result of the fitting procedure.

To test whether DM performance predictions can be robustly reproduced we divided all subjects into six groups according to PMAT24_A_RTCR and fitted each 100 times, randomizing seeds and initial conditions as above. In all 100 tests mean amplitudes decreased and correlations increased from low to high PMAT24_A_RTCR (Supplementary Fig. 8; Friedman test rejected the null hypothesis that distributions are equal with $p = 0$; post-hoc multiple comparison analysis using Nemenyi's test showed that the six groups were significantly different with $p < 0.001$ for all pairs), confirming that the identified link to empirical performance is a general result of the fitting procedure.

To test whether there is a robust relationship and comparability between inferred synaptic inputs across the subject population we trained regression models on one half of the cohort and then applied the model on the second half to estimate its generalizability and repeated this process 1000 times to obtain a statistic over different random train and test groups. Predicting subject-wise mean FC from mean synaptic inputs yielded correlations of $r = 0.67 \pm 0.025$ for the training sets and $r = 0.66 \pm 0.025$ for the test sets. Next, using the mean inputs from the ten areas with highest correlation with mean FC as independent variables yielded a fit of $r = 0.79 \pm 0.018$ with the training sets and $r = 0.73 +/-0.055$ with the test sets. Lastly, we computed regression models for every single FC connection ($N = 71,631$) using the mean input currents from the ten areas with highest correlation with the respective FC connection as independent variables. Over all connections, this yielded an average fit of $r = 0.61 +/-0.1$ for the training and $r = 0.52 +/-0.13$ for the test set. The stability of prediction qualities in test versus train sets in above tests indicates that the inferred properties are meaningfully comparable across the subject population.

## Discussion

We propose that FC and synchrony between brain areas directly depend on the ratio of their mutual excitation and inhibition. This theoretical observation yielded a parameter optimization algorithm that enabled to fit whole-brain simulated FCs to their empirical counterparts based on a Hebbian learning rule that implements homeostatic plasticity of excitation-inhibition balance in brain network models[42]. The dynamics of the resulting $N = 650$ models were then linked with the subjects' empirical intelligence test scores and used to explain individual differences in cognitive performance. The research yields an implementation of multiscale brain network models that are able to perform decision-making tasks, both of which have recently been identified as crucial steps to explain the relationship between microscopic phenomena, large-scale brain function, and behavior as well as generating brain digital twins for personalized medical interventions[43]. The obtained insights held true independent of any parameter fitting in subsequent tests with isolated circuits. In addition, strict tests were employed to ensure the generality of the fitting procedure. Although we here focus on individual variability in DM and intelligence of healthy individuals, the insight that E/I-balance can be

used to precisely set FC in brain models indicates a general-purpose method for inferring healthy and pathological neural mechanisms underlying functional brain networks. This is particularly relevant for clinical applications as impaired E/I-balance has now become a refined framework for understanding neurological diseases including autism spectrum disorders, schizophrenia, neurodegenerative diseases, and neuropsychiatric disorders[42,44].

It must be mentioned as a limitation that BNMs are high-dimensional models with thousands of parameters and the identified mechanism may be one out of a potentially infinite number of mechanisms that could explain the observed data. As with any scientific hypothesis, it is therefore crucial to validate and falsify theory with dedicated experiments. Since the used brain network model simulates detailed properties of neural systems like input currents, firing rates, synaptic activity, and fMRI, it is directly amenable for further validation or falsification with empirical data from different modalities. By integrating diverse empirical findings into a unifying computational framework that can be iteratively refined (or refuted) dynamic models provide an avenue out of the 'reproducibility crisis'[7]. BNMs are limited when it comes to their resolution, as they are typically based on connectivity data obtained from non-invasive imaging techniques like MRI and limited computational power to simulate large networks. These problems are addressed with multiscale models where only some parts of the brain are simulated at a finer scale (for example, at the level of spiking neurons[45]) while the remaining parts are simulated by a coarser network to save computational resources. In addition, by integrating connectivity and other microstructural information from finer scale studies, for example, from invasive rodent studies[46] or post-mortem human atlases[47], it becomes possible to further constrain parameters and test the plausibility of simulation results. In this regard, we note that the described relationship between E/I-balance and FC (respectively population synchronization) appears independent of the spatial and temporal scales of the network, and may be used to generally tune also finer-scale or coarser-scale networks, as it is based on generic dynamical primitives of neural mass action applicable to describe dynamics across spatial and temporal scales[48]. Although BNMs employ abstractions, like all models, further advances may emerge precisely where the assumptions break down. For example, the used ensemble models capture neural population dynamics primarily when coherence is sufficiently weak that individual spikes can be ignored or when coherence is sufficiently strong that variance can be considered small, while scale-free dynamics with unbounded variance resist mean-field reductions and may require alternative ensemble methods[7,8]. Despite these limitations, BNMs are in contrast with artificial neural networks specifically designed to explain the underlying biology, using typically observed features of the empirical system as targets for validation and falsification (Supplementary Fig. 10) to achieve an incrementally improved computer model of the empirical system.

In this work, we found that DM accuracy can be traded with DM speed in dependence of brain network configuration. Faster is therefore not necessarily better, but rather the ability to switch between fast and deep modes of information processing—depending on the nature of the problem and the involved brain areas. The idea that decision-making speed is traded with accuracy is supported by numerous empirical findings in the fields of economy, ecology, psychology, and neuroscience[9,49–51].

Our modeling results now cast this idea in terms of neural network interaction: FC depends on E/I-ratios, E/I-ratios modulate synaptic inputs, which in turn modulates evidence integration in winner-take-all circuits. Decreased synaptic inputs prolong the time window for integration and make DM more dependent on the buildup of slowly reverberating activity between PFC and PPC regions, pointing to a general mechanism that gives higher-order populations top-down control. Slowing down the timescale may bring DM under conscious control, enabling to modulate DM by attentional processes, which is supported by empirical results that associate top-down attention with amplification of PPC activity and increased correlation between PFC and PPC[52]. This idea was formulated as the 'ignition' theory of conscious processing, stating that while most of the brain's early computations can be performed in a non-conscious mode, conscious perception is associated with long-distance integration of activity in frontoparietal circuits[52,53]. Importantly, the specific markers that contrast conscious from nonconscious processing overlap with those needed for DM slowing in our model. In the experimental literature conscious perception is systematically associated with surges of prefrontal activity followed by top-down parietal amplification: conscious access crucially depends on a sudden, late, all-or-none ignition of prefronto-parietal networks and subsequent amplification of sensory activity[54]. The most consistent correlate of conscious perception was a late (-300–500 ms) positive waveform in prefrontal regions that reactivated parietal regions along with increased long-distance synchronization in the frontoparietal network[54], which strongly resembles our model's behavior: in the slow DM mode ramping of PFC was necessary to amplify activity in PPC, while in the fast DM mode PPC ramping preceded the ramping of prefrontal cortex (Supplementary Figs. 4 and 5 and Supplementary Movie 1). Similarly, monkey recordings showed that WM content in PFC neurons was multiplexed with signals that reflected the subject's covert attention[55]. Together with the observation that subjects' performance and WM load correlated with the degree of prefronto-parietal synchronization[56,57], the conclusion can be drawn that these processes may reflect top-down prefrontal attentional mechanisms that modulate processing in posterior cortex. Likewise, these results also integrate with the parieto-frontal integration theory of intelligence, which roughly states that after basic processing in temporal and occipital lobes, sensory information is collected in parietal cortex, which then interacts with frontal regions to perform hypothesis tests on attempted solutions to select an optimal solution[58].

Another relevant perspective is provided by the distinction into effortful versus automatic cognition: while effortful tasks require synchronization or parietal regions with PFC, the synchronization suddenly drops as soon as subjects move into a routine mode of task execution[59,60]. Similarly, harder decisions required slow integration in the model's PFC-PPC network, while simpler decisions were quickly taken by the PPC module. A number of FC studies come to similar conclusions: FC became more integrated during challenging tasks and remained more segregated during simple tasks[61–65]. Likewise, work on short-term synaptic plasticity suggests that FC is changed to form temporary task-relevant circuits, which comes with energetic and computational advantages[66], similar to the influential Communication through Coherence theory[67], which proposes phase synchronization as an essential and generic mechanism for controlling selective information flow in multiplexed brain networks.

More generally, our study indicates that areas with higher FC may interact on a slower time scale than areas with lower FC. These different time scales could give rise to a hierarchical information processing cascade where intermediate results from faster processes are integrated by slower processes, which is reflected in the emerging view that cortex posits a timescale-based topography with integration windows increasing from sensory to association areas[68]. As receptive windows are progressively enlarged along the hierarchy, DM integration is extended from local to long-range circuits integrating increasingly widespread information, which is supported by studies that show how slow (<0.1 Hz) power fluctuations reliably track the accumulation of complex sensory inputs in higher-order regions[69].

Summarizing, in the present work we identified a monotonic and smooth relationship between the structural and the functional architecture of neural networks: by tuning the E/I-ratio it became possible to precisely and simultaneously tune the FC between any pair of network nodes to the desired target configuration from full

antisynchronization to full synchronization. We believe this is important, as the link between FC and structural brain architecture is often described as unclear and many research streams aim for inferring structural network topology[70,71]. We therefore expect that the described smooth and monotonic link between network architecture and FC, and the derived learning rule, will be useful to better understand and infer structural network mechanisms underlying healthy and pathological cognition[72,73].

## Methods

### Large-scale brain network model

The used large-scale BNM simulates brain activity based on the network interaction of population models that represent brain areas. Each brain area is simulated by coupled excitatory and inhibitory population models based on the dynamical mean field model, which was derived from a detailed spiking neuronal network model[32,74,75]. Populations are connected by structural connectomes estimated from dwMRI data via fiber tractography. Here, we extended the model using two additional parameters $w_{ij}^{LRE}$ and $w_{ij}^{FFI}$ that allow the balancing of long-range excitatory and feedforward inhibitory synaptic currents. The model equations read as follows.

$$I_i^E = W_E I_0 + w_+ J_{NMDA} S_i^E + J_{NMDA} \sum_j w_{ij}^{LRE} C_{ij} S_j^E - J_i S_i^I \quad (1)$$

$$I_i^I = W_I I_0 + J_{NMDA} S_i^E + J_{NMDA} \sum_j w_{ij}^{FFI} C_{ij} S_j^E - S_i^I \quad (2)$$

$$r_i^E = \frac{a_E I_i^E - b_E}{1 - \exp(-d_E(a_E I_i^E - b_E))} \quad (3)$$

$$r_i^I = \frac{a_I I_i^I - b_I}{1 - \exp(-d_I(a_I I_i^I - b_I))} \quad (4)$$

$$\frac{dS_i^E(t)}{dt} = -\frac{S_i^E}{\tau_E} + (1 - S_i^E)\gamma_E r_i^E + \sigma v_i(t) \quad (5)$$

$$\frac{dS_i^I(t)}{dt} = -\frac{S_i^I}{\tau_I} + \gamma_I r_i^I + \sigma v_i(t) \quad (6)$$

$r_i^{(E,I)}$ denotes the population firing rate of the excitatory ($E$) and inhibitory ($I$) population of brain area $i$. $S_i^{(E,I)}$ identifies the average excitatory, respectively inhibitory, synaptic gating activity of each brain area. The sum of all input currents to each area are identified by $I_i^{(E,I)}$ (units nA). $W_{(E,I)} I_0$ are the overall effective external input currents to excitatory, respectively inhibitory, populations, and $w_+$ the local excitatory recurrence. $J_{NMDA}$ and $J_i$ are parameters that quantify the strengths of excitatory synaptic coupling and local feedback inhibitory synaptic coupling, respectively. Feedback inhibition control using inhibitory synaptic plasticity modulates $J_i$ of each region such that the long-term average firing rate $r_i^E$ of the corresponding excitatory population is ~4 Hz (see section Feedback Inhibition Control). We extended the original model by Deco et al.[32]. by introducing the parameters $w_{ij}^{LRE}$ and $w_{ij}^{FFI}$, which are matrices with the same dimensions as the structural connectome $C_{ij}$ (regions × regions) that describe the strengths of long-range excitation and feedforward inhibition, respectively. Equations 3 and 4 are sigmoidal functions that convert input currents into firing rates. $\tau_{(E,I)}$ and $\gamma_{(E,I)}$ specify the time scales and rate of saturation of excitatory and inhibitory synaptic activity, respectively. $v_i(t)$ is noise drawn from the standard normal distribution. Table 1 lists all state variables as well as parameters and their settings. BOLD activity was simulated by inputting excitatory

synaptic activity $S_i^E$ into the Balloon-Windkessel hemodynamic model[76], which is a dynamical system that describes the transduction of neuronal activity into perfusion changes and the coupling of perfusion to BOLD signal. The model is based on the assumption that the BOLD signal is a static non-linear function of the normalized total deoxyhemoglobin voxel content, normalized venous volume, resting net oxygen extraction fraction by the capillary bed, and resting blood volume fraction. Please refer to Deco et al.[74] for the specific set of Ballon−Windkessel model equations that we used in this study.

### Multi-scale brain network model

To form the multiscale model, we connected the two-module DM circuit functional WM/DM circuit[33] to the large-scale regions that simulate PPC and PFC. To connect the large-scale network with the mesoscopic network, we augmented the noise terms of the DM circuit network by large-scale BNM inputs to PPC and PFC. The equations of the DM circuit read as follows.

$$I_i^n = \sum_{m,j} S_j^m J_{i,j}^{m \to n} + I_0 + I_{app,i}^n + I_{noise,i}^n \quad (7a)$$

$$I_i^n = \sum_{m,j} S_j^m J_{i,j}^{m \to n} + I_0 + I_{app,i}^n + I_{BNM,i}^n \quad (7b)$$

$$r(I) = \frac{aI - b}{1 - \exp(-c(aI - b))} \quad (8)$$

$$\frac{dS_i^n}{dt} = -\frac{S_i^n}{\tau} + \gamma(1 - S_i^n) r(I_i^n) \quad (9)$$

$$\tau_{AMPA} \frac{dI_{noise,i}^n(t)}{dt} = -I_{noise,i}^n(t) + \eta_i(t)\sqrt{\tau_{AMPA} \sigma_{noise}^2} \quad (10)$$

Parameter values and state variables have the corresponding meanings as in the equations for the large-scale models (see also Supplementary Table 2 for an overview of quantities and values). Equation 7a shows the original DM circuit model input equation with noise term $I_{noise,i}^n$. To couple the DM circuit to the large-scale network, the noise term $I_{noise,i}^n$ was replaced by the term $I_{BNM,i}^n$ in Eq. 7b. The term adds the noise process from the DM circuit model (Eq. 10) to the large-scale BNM input to drive the DM circuit:

$$I_{BNM,i}^n = (b_{MJW} - a_{MJW})[(w_+ J_{NMDA} S_i^E + J_{NMDA} \sum_j w_{ij}^{LRE} C_{ij} S_j^E - J_i S_i^I)$$
$$- a_{BNM,i}]/(b_{BNM,i} - a_{BNM,i}) + a_{MJW} + I_{noise,i}^n(t) \quad (11)$$

Similar to Eq. 1 the input from the BNM to the DM circuit populations consists of the sum of local recurrent excitation, global network input and local recurrent inhibition. For each region this input is range normalized to bring the range of amplitudes from the 650 individual models into a range suitable for the DM circuit as identified in Fig. 4, with $b_{MJW} = 0.001(nA)$ and $a_{MJW} = -0.006(nA)$. For each region the amplitude ranges from the 10th percentile to the 90th percentile over the 650 BNMs was mapped into the range $[a_{MJW}, b_{MJW}]$. To the resulting amplitude the individual Ornstein−Uhlenbeck noise processes were added in order to make one variant of this input for each of the two nodes of one DM circuit module.

Decision-making performance was computed as in the original publication of the DM circuit by Murray et al. by modeling the strength of evidence as an external current to the two parietal populations $A_{PPC}$ and $B_{PPC}$ as follows:

$$I_{app,i}^n = I_e \left(1 \pm \frac{c'}{100\%}\right) \quad (12)$$

where $I_e = 0.0118$ nA scales the overall strength of the input and $c' = 6.4\%$, referred to as the strength of evidence or contrast, determines which of the two populations $A_{PPC}$ or $B_{PPC}$ receives higher evidence ($A_{PPC}$ received the higher evidence), which reflects the saliency of the target with respect to that of distractors. As in Murray et al.[33], when one of the two action populations $A_{PFC}$ or $B_{PFC}$ reaches a firing rate threshold of 40 Hz the decision for option A or B is taken and a reaction time is registered. We repeated the decision-making task 1000 times in order to compute the percentage of times for which the decision was made correctly (number of times $A_{PFC}$ crossed the firing-rate threshold divided by the total number of trials) and the average time until the threshold was reached.

## Fitting algorithm

The fitting algorithm is based on the observation (Fig. 2) that the correlation between the fMRI time series from two different model brain regions can be modulated by the relative strengths of long-range excitation versus feedforward inhibition. This ratio, as outlined in the section Large-scale brain network model, can be adjusted in our model by the parameters $w_{ij}^{LRE}$ and $w_{ij}^{FFI}$, which are multiplicative factors that re-scale the structural connectivity connection weights $C_{ij}$, between each pair of connected regions $i$ and $j$. $w_{ij}^{LRE}$ modulates the amount of excitation conveyed via long-range connections to distant excitatory populations, or long-range excitation, while $w_{ij}^{FFI}$ modulates the amount of excitation provided via long-range connections to distant inhibitory populations, and their resulting feedforward inhibitory effect on the accompanying excitatory population, or feedforward inhibition. The goal of the fitting algorithm is to fit weights $w_{ij}^{LRE}$ and $w_{ij}^{FFI}$ such that FC computed from simulated fMRI time series matches a target FC as closely as possible. The goal is that the difference between each entry $\rho_{ij}^{trg}$ of the target FC matrix $\boldsymbol{\rho}^{\mathbf{trg}}$ and $\rho_{ij}^{sim}$ of the simulated FC matrix $\boldsymbol{\rho}^{\mathbf{sim}}$ should be as small as possible. The basic idea of the fitting algorithm is to increase $w_{ij}^{LRE}$ and to decrease $w_{ij}^{FFI}$ if $\rho_{ij}^{trg} > \rho_{ij}^{sim}$ and, vice versa, to decrease $w_{ij}^{LRE}$ and to increase $w_{ij}^{FFI}$ if $\rho_{ij}^{trg} < \rho_{ij}^{sim}$. While the overall parameter optimization approach followed a standard gradient descent schema, importantly, the gradients are based on the direct monotonic and smooth relationship that we identified between E/I-ratios and FC, respectively population synchronization (Fig. 4), creating a direct biologically interpretable link between brain network structure (specifically the E/I-ratios between network nodes) and the emerging brain network dynamics when simulating the model. In pseudocode the algorithm can be written as follows.

**Algorithm. EI_tuning($\boldsymbol{\rho}^{\mathbf{trg}}$, $\eta$, M)**
 **Input** $\boldsymbol{\rho}^{\mathbf{trg}}$: (n x n) target FC matrix
 $\eta_{EI}$: scalar learning rate
 M: brain network model
 **Returns** $w^{LRE}$, $w^{FFI}$: (n x n) matrices long-range excitation, feed-forward inhibition
 **for** fmri_time_step = 1 to simulation_length **do**
 simulate one fMRI time step using M
 compute simulated FC $\boldsymbol{\rho}^{\mathbf{sim}}$
 **for** i = 1 **to** n **do**
 rmse_i = root-mean-square deviation between matrix rows i in $\boldsymbol{\rho}^{\mathbf{trg}}$ and $\boldsymbol{\rho}^{\mathbf{sim}}$:
 **for** j = 1 to number of connections of node i **do**
 diff_FC = $\rho_{ij}^{trg}$ - $\rho_{ij}^{sim}$
 $w_{ij}^{LRE}$ = $w_{ij}^{LRE}$ + $\eta_{EI}$ x diff_FC x rmse_i
 $w_{ij}^{FFI}$ = $w_{ij}^{FFI}$ - $\eta_{EI}$ x diff_FC x rmse_i
 **if** $w_{ij}^{LRE} \leq 0$ **do** $w_{ij}^{LRE} = 0$
 **if** $w_{ij}^{FFI} \leq 0$ **do** $w_{ij}^{FFI} = 0$
 return $w^{LRE}$, $w^{FFI}$

The algorithm iterates over all connections $(i,j)$ of the BNM and computes the difference between target and simulated FC for each connection. This difference is rescaled by the learning rate $\eta$, which is gradually decreased over the course of the tuning. Furthermore, the difference is re-scaled by the root-mean-square deviation (RMSE) between the correlation coefficient values of region $i$ with all remaining regions (i.e. the RMSE between rows $i$ of matrices $\boldsymbol{\rho}^{\mathbf{trg}}$ and $\boldsymbol{\rho}^{\mathbf{sim}}$), which can be compared to the temperature parameter in a simulated annealing heuristic. The factor has the purpose to decrease the change in $w_{ij}^{LRE}$ and $w_{ij}^{FFI}$ as the fit of the row-wise FC increases and we approach an optimum. Furthermore, the factor differentially weights the changes in $w_{ij}^{LRE}$ ($w_{ij}^{FFI}$) and $w_{ji}^{LRE}$ ($w_{ji}^{FFI}$) with the purpose that the region (either $i$ or $j$) that has a better fit at the current tuning iteration is changed less than the other one, since the change of connection strengths between one region pair has an effect on the FC between all other region pairs. By decreasing the step size for the better-fitting region, we ensure that respective parameters stay closer to the local optimum. In the present paper this heuristic is used as an online parameter tuning rule, which means that parameters are updated after each new BOLD fMRI time step is computed. We tested different values for the learning rate parameter $\eta_{EI}$, and devised a tuning workflow in which initially the parameter space is sampled with large steps (large learning rate) using FC that is based on a short time window. The tuning lasted over six stages where each stage was simulated for 10 hours of biological time. The learning rate $\eta_{EI}$ was halved and the window size of simulated FC $\boldsymbol{\rho}^{\mathbf{sim}}$ was doubled in each stage, starting with a learning rate of $\eta_{EI} = 0.1$ and a window size of 150 TRs. The wall time for simulating one hour of biological activity on one standard CPU was roughly two hours, which led to a computational cost of 6[Stages]*20[CPU hours per stage] = 120[CPU hours] to tune a single model and a total cost of 78,000 CPU hours to tune all 650 models. Fitting runs were executed in parallel on high performance computers. The costs for running subsequent DM and WM experiments with the fitted and coupled multiscale models were negligible and performed on a standard laptop as only a few seconds of activity were needed to simulate one DM or WM experiment.

## Feedback inhibition control

The firing rate of the large-scale neural masses (Eqs. 3 and 4) depends on synaptic input currents (Eqs. 1 and 2), which are, to a large degree, determined by the structural connectome $C$, that is, large-scale inputs, and associated parameters ($w_{ij}^{LRE}$ and $w_{ij}^{FFI}$). To compensate for excess or lack of excitation, which would result in implausible firing rates, a local regulation mechanism, called feedback inhibition control (FIC), was used. The approach was previously successfully used to significantly improve FC prediction, and for increasing the dynamical repertoire of evoked activity and the accuracy of external stimulus encoding[32,77]. To implement FIC we used a learning rule for inhibitory synaptic plasticity that was shown to balance excitation and inhibition in sensory pathways and memory networks[31]. The learning rule modulated all connection strengths from inhibitory to local excitatory populations once every 720 ms (corresponding to 1 fMRI repetition time) to achieve a target average firing rate of 4 Hz in excitatory populations. The learning rule can be summarized as

$$\triangle w = \eta_{FIC}(\text{pre} \times \text{post} - \rho_0 \times \text{pre}) \qquad (13)$$

where $\triangle w$ denotes the change in synaptic strength, pre and post are the pre- and postsynaptic firing rates, $\eta_{FIC} = 0.001$ is the learning rate and $\rho_0 = 4.0[Hz]$ is the target firing rate for the postsynaptic excitatory population. If postsynaptic firing rate *post* is larger than the target firing rate $\rho_0$, the learning rule increases the inhibitory weight $w$, to decrease the postsynaptic firing rate. Conversely, if the postsynaptic firing rate is lower than the target firing rate, the learning rule decreases the inhibitory weight. The change of the inhibitory weight is modulated by the presynaptic firing rate pre: if presynaptic firing is large, then a higher weight change is needed to get the desired effect than when presynaptic firing is low. The learning rate $\eta$ was found by trial and error.

## Data and preprocessing

We used the publicly available HCP Young Adult data release[16], which includes behavioral and 3 T MR imaging data from healthy adult participants (age range 22–35 years). Informed consent forms, including consent to share deidentified data, were collected for all subjects (within the HCP) and approved. Data collection was approved by a consortium of institutional review boards in the United States and Europe, led by Washington University (St Louis) and the University of Minnesota (WU-Minn HCP Consortium). The experiments were performed in compliance with the relevant laws and institutional guidelines and were approved by the medical ethical committee of the Charité Medical Center in Berlin (EA4/184/20). All data were collected on a 3 T Siemens Skyra scanner with gradients customized for the HCP. We restricted our analysis to 650 subjects (360 female, 290 male, based on self-report during data collection by the HCP; no analyses regarding sex or gender were performed as the goal of this study was to elucidate mechanisms that are independent of sex or gender) with complete MRI data including all four sessions of resting-state fMRI, structural MRI (T1w and T2w), diffusion-weighted MRI (dwMRI) as well as the behavioral measures PMAT24_A, CardSort, ProcSpeed and Flanker were available, and which were not identified with quality issues by HCP. The HCP publishes lists with subjects where quality control issues were identified (https://wiki.humanconnectome.org/pages/viewpage.action?pageId=88901591), which involved 151 subjects at the time of writing. Furthermore, we identified one additional subject that had absent connections that was more than four standard deviations away from the mean over all subjects. Resting-state fMRI data were acquired in four separate 15-min runs on two different days (two per day) with a 2-mm isotropic spatial resolution (FOV: 208 mm × 180 mm, Matrix: 104 × 90 with 72 slices covering the entire brain) and a 0.73-s temporal resolution. For correction of EPI distortions, additionally two spin echo EPI images with reversed phase encoding directions were acquired. dwMRI had a resolution of 1.25 mm isotropic, 128 diffusion gradient directions, and multiple q-space shells with diffusion-weightings of $b = 1000$ s/mm2, $b = 2000$ s/mm$^2$ and $b = 3000$ s/mm$^2$. For correction for EPI and eddy-current-induced distortions two phase-encoding direction-reversed images for each diffusion direction were acquired. From HCP, we downloaded preprocessed fMRI, structural MRI and dwMRI data that underwent HCP's preprocessing pipelines, which combine tools from FSL, FreeSurfer and the HCP Connectome workbench to perform distortion correction and alignment across modalities[78]. For high-resolution (0.7-mm isotropic) T1-weighted and T2-weighted MR scans HCP pipelines corrected for distortions using a B0 field map and then linearly registered the anatomy with a common MNI template. Individual surface registration was achieved by combining cortical surface features and a multimodal surface matching algorithm[79]. fMRI pipelines include distortion-correction, motion correction, registering fMRI data to structural data, reduction of the bias field, normalization to a global mean, brain masking, re-sampling of fMRI time series from the volume into the gray-ordinates standard space, and denoising using FSL's ICA-FIX method. Corrected time series were then sampled into HCP's 91,282 standard grayordinates (CIFTI) space, which is a combined representation of a cortical surface triangulation (32k vertices per hemisphere) and a standard 2 mm subcortical segmentation[78]. We parcellated grayordinate fMRI time series using HCP's multimodal parcellation[41] and computed region-wise average time series and FC matrices. For dwMRI data, HCP pipelines normalize the b0 image intensity across runs; remove EPI distortions, eddy-current-induced distortions, and subject motion; correct for gradient-nonlinearities; perform registration with structural data, resamples into 1.25 mm structural space; and mask the data with a brain mask. For dwMRI tractography we employed our own pipelines[80] based on the tractography toolbox MRtrix3[81]. Structural MRI images were segmented into five tissue types to aid Anatomically-Constrained Tractography,

a MRtrix3 function that removes anatomically implausible tracks. Multi-shell, multi-tissue response functions were estimated using MRtrix3 software dwi2response, followed by multi-Shell, Multi-Tissue Constrained Spherical Deconvolution using dwi2fod. For each subject full-brain tractograms with 25 Million tracks were generated using tckgen, subsequently filtered with tcksift2, and mapped to the HCP MMP parcellation used for computing fMRI FC to produce matching structural connectomes. The g-factor was computed using the code of Dubois et al. who performed factor analysis of the scores on 10 cognitive tasks from the HCP data set to derive a bi-factor model of intelligence, which is the standard in the field of intelligence research[82].

## Statistical tests

To test whether simulated data samples that we obtained for the different RT groups have the same or different distributions we used the nonparametric Friedman test (implemented by the function friedmanchisquare() in the Python package SciPy stats) followed by a posthoc multiple comparison analysis using Nemenyi's test (using the function posthoc_nemenyi_friedman() implemented in the Python package scikit-posthocs). Data samples were not normally distributed (tested with Lilliefors test) and contained repeated measurements (each group model was fitted 500 times with different initial conditions and then simulated). To test whether medians are equal for data with unequal sample sizes and without repeated measurements we used the Kruskal–Wallis test followed by posthoc Conover's test (implemented as SciPy functions kruskal() and posthoc_conover()) for pairwise multiple comparisons.

## Reporting summary

Further information on research design is available in the Nature Portfolio Reporting Summary linked to this article.

## Data availability

All data used in this study was derived from the Human Connectome Project Young Adult study available in the repository https://db.humanconnectome.org/data/projects/HCP_1200. The derived data generated in this study are available under restricted access due to data privacy laws, access can be obtained within a timeframe of one month from the corresponding authors M.S. and P.R. as processing and sharing is subject to the European Union General Data Protection Regulation (GDPR), requiring a written data processing agreement, involving the relevant local data protection authorities, for compliance with the standard contractual clauses by the European Commission for the processing of personal data under GDPR (https://commission.europa.eu/publications/standard-contractual-clauses-controllers-and-processors-eueea_en). The data processing agreement and dataset metadata are available in EBRAINS (https://search.kg.ebrains.eu/instances/88507924-8509-419f-8900-109accf1414b).

## Code availability

All custom codes used in this study are freely available at GitHub (https://github.com/BrainModes/fast-slow/)[83] licensed under the EUPL-1.2-or-later. Custom codes were implemented using Python version 3.9.7 and multiple Python packages (scipy 1.7.1; numpy 1.20.3; matplotlib 3.4.3; scikit-learn 1.1.3; statsmodels 0.12.2; scikit-posthocs 0.7.0) and MATLAB version R2020a; GCC 9.4 was used for C code compilation; FreeSurfer v7.1.0, MRtrix3 3.0, FSL 6.0 for MRI processing.

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

## Acknowledgements

We gratefully acknowledge the Swiss National Supercomputing Center CSCS for supporting this project by providing computing time through the Interactive Computing E-Infrastructure (ICEI) on the Supercomputer Piz Daint of the Fenix Infrastructure (projects ich10, ich12) and the Gauss Centre for Supercomputing e.V. (www.gauss-centre.eu) for supporting this project by providing computing time through the John von Neumann Institute for Computing (NIC) on the GCS Supercomputer JUWELS at Jülich Supercomputing Centre (JSC). Data were provided in part by the Human Connectome Project, WU-Minn Consortium (Principal Investigators: David Van Essen and Kamil Ugurbil; 1U54MH091657) funded by the 16 NIH Institutes and Centers that support the NIH Blueprint for Neuroscience Research; and by the McDonnell Center for Systems Neuroscience at Washington University. This work was supported by the Virtual Research Environment at the Charité Berlin – a node of EBRAINS Health Data Cloud. We gratefully acknowledge support by Digital Europe Grant TEF-Health # 101100700 (P.R.); H2020 Research and Innovation Action Grant Human Brain Project SGA2 785907 (P.R.); H2020 Research and Innovation Action Grant Human Brain Project SGA3 945539 (P.R.); H2020 Research and Innovation Action Grant Interactive Computing E-Infrastructure for the Human Brain Project ICEI 800858 (P.R.); H2020 Research and Innovation Action Grant EOSC VirtualBrainCloud 826421 (P.R.); H2020 Research and Innovation Action Grant AISN 101057655 (P.R.); H2020 Research Infrastructures Grant EBRAINS-PREP 101079717 (P.R.); H2020 European Innovation Council PHRASE 101058240 (P.R.); H2020 Research Infrastructures Grant EBRAIN-Health 101058516 (P.R.); H2020 European Research Council Grant ERC BrainModes 683049 (P.R.); JPND ERA PerMed PatternCog 2522FSB904 (P.R.); Berlin Institute of Health & Foundation Charité (P.R.); Johanna Quandt

Excellence Initiative (P.R.); German Research Foundation SFB 1436 (project ID 425899996) (P.R.); German Research Foundation SFB 1315 (project ID 327654276) (P.R.); German Research Foundation SFB 936 (project ID 178316478) (P.R.); German Research Foundation SFB-TRR 295 (project ID 424778381) (P.R.); German Research Foundation SPP Computational Connectomics RI 2073/6-1, RI 2073/10-2, RI 2073/9-1 (P.R.).

## Author contributions

Conceptualization: M.S., P.R. Methodology: M.S., G.D., P.R. Software: M.S. Validation: M.S., G.D., P.R. Formal analysis: M.S., P.R. Writing: M.S., P.R. Visualization: M.S., P.R. Funding acquisition: P.R.

## Funding

## Competing interests

The authors M.S. and P.R. declare the following competing interests: the authors M.S. and P.R. are filing a patent application entitled Tuning a Biological Network Model (227EP 3414) for the fitting algorithm described in this paper. The remaining author declares no competing interests.
