## [Peer Review File · Nature Communications]

REVIEWER COMMENTS

Reviewer #2 (Remarks to the Author):

Summary

The authors in this work have analyzed data from the Human Connectome Project regarding correlations between general and fluid intelligence and fast and slow decision-making. The analyses revealed that individuals with higher general and fluid intelligence made easy decisions faster but took more time for complex problems. The authors then developed a computational model based on mean-field approximations and the results of their analysis on the HCP data. The authors proposed a fitting algorithm for tuning the E/I ratios within their model as well as a learning rule for feedback inhibition control. The model predicts a mechanistic link between excitation-inhibition balance, fluid intelligence, decision-making, working memory, reaction times, and brain synchrony. Furthermore, the model manifests the same trade-off between speed and accuracy as the human subjects in the Human Connectome Project.

The manuscript is well-organized and well-written. The authors can improve the figures and add more information in the captions (see my comments below). The numerical results support the authors' claims, and their model correctly predicts the empirical results. The references in this work are adequate.

Comments

1. The authors estimated correlations between at most two variables, for instance, the g-factor vs. reaction times (RT) or functional connectivity (FC) and PMAT24_A_CR. What happens when one measures the correlation between multiple variables, such as g-factor and RT vs. FC? In other words, how do all the variables reported in the text affect FC?
2. In figure 2c, I assume "original" means the model without tuning. Please clarify that in the text. Furthermore, in figure 2b, what does "lower triangular" mean? The matrix in the bottom panel of 2b does not look lower triangular.
3. Please add the corresponding panels of figure 3 in lines 207 - 211, where you present the results on how amplitude and synaptic inputs affect the performance.

4. It is unclear why the authors devised their fitting algorithm instead of a conventional one to find the optimal values for W_{LRE} and W_{FFI} .

5. The authors should provide more details on the learning rate schedule (Was it linear, exponential, or something else?).

Minor Comments

1. The text's abbreviations in Figure 1, Tables 1 and 2, and elsewhere are confusing.

2. The captions of the figures need some improvement. The figures, in some cases, are not self-explanatory. For instance, there needs to be more information regarding the colormaps. The histograms are hard to read. The same is true for supplementary figures (for example, S3). Please ensure that the captions provide all the necessary information for the reader to understand the content of the figures without going back and forth in the main text.

3. In table 2, where the authors refer to PMAT24_A_RTCT, I assume that the reference number is 15 (third column).

4. The section title in line 169 seems confusing.

5. Please comment on the computational cost of your algorithms and your simulations in general. How long does it take to run one experiment?

6. The authors use the same symbol for the learning rate of the fitting algorithm and the feedback inhibition rule (η). It might be obscure to some readers, so please use different symbols.

7. Line 473 typo (η not ϵ).

8. The Github link to the source code is just a placeholder. Please add the source code.

Reviewer #3 (Remarks to the Author):

In this paper, Schirner and colleagues use whole-brain modeling fit to individual subjects' fMRI data to investigate the mechanistic basis of fluid intelligence during complex cognitive tasks. Their main results show that:

- there exists non-trivial correlations between reaction times and general measures of intelligence (g-factor): negative correlation for easier problems (higher intelligence \rightarrow faster problem solving), and

positive correlation for harder problems; while there are varying degrees of correlation between resting state functional connectivity (rsFC) and task measures / intelligence, most notably, task reaction time

- using subject-specific rsFC as the target, DTI-derived structural connectivity as the connectivity matrix, and the ratio between long-range excitation (LRE) and feedforward inhibition (FFI) as free parameters, the authors can tune individualized, whole-brain, 379-node neural mass models with local E/I populations such that the simulated fMRI data exhibits similar rsFC patterns for all subjects (in terms of correlation and RMSE)
- there are correlations between task measures (at the individual and stratified levels), such as RT, and the simulated dynamics of the individualized brain models, in particular mean synaptic amplitude and correlation
- in a 2-node model of PFC-PPC interaction, the authors identify synaptic input amplitude and coupling degree (correlation) as important parameters for task performance and speed, in a nonlinear fashion; they then insert the 2-node model into the whole-brain model such that the simulated synaptic currents now drive the 2-node model, and find that simulated performance and speed match empirical data at the stratified level
- they further validate the computational findings by refitting the model many times, observing similar simulated time series and directions of correlation despite apparent variance in fitted parameter values

I think this is an interesting paper that aims to combine neurophysiological mechanisms, computational modeling of brain dynamics and functional connectivity, and cognition. If I'm not mistaken, the main contributions can be summarized as "personalized virtual brain model reproduces subject-specific rsFC patterns, as well as nonlinear relationship between speed and performance in fluid intelligence tasks (as well as the relationship between neural and behavior data), and suggests aspects of synaptic dynamics, driven by the balance between long-range E and I strengths, as important contributing factors." In my opinion, this type of work that bridges neural mechanisms and complex cognition is valuable for neuroscience, even if obvious abstractions on the modeling side are necessary. Additionally, I particularly appreciated the "model validation" section, as ensuring the robustness of results in a computational work is crucial.

On the other hand, I have numerous major concerns regarding the framing of the results, in particular the mismatch between the claimed significance and the actual results (and a lack of acknowledgement there), and the many ambiguities surrounding crucial technical details (expanded below). These issues are compounded by what I personally found to be a difficult presentation, both in terms of the organization of the narrative at a high level, as well as a lack of key main figures and descriptive figure captions. I apologize if I had misinterpreted any of the results in my summary, I simply could not find a lot of information in the main text and methods, but if the above are accurate in the authors' eyes, then I think the paper could be improved on the presentation front in order to make it easier for a reader to digest the main messages. As a whole, I found the paper to have some interesting results and refreshing approaches, but is extremely over-interpreted, lacking in crucial methodological details, and presented

in a very confusing way. I outline my major concerns below, from which I hope the authors can derive some helpful suggestions.

Concerns regarding presentation and framing:

I appreciate that the authors are undertaking a difficult task in framing a paper which encapsulates results across neuroimaging, computational modeling, and behavioral analysis. Nevertheless, the paper currently reads as if it was written for a condensed Nature or Science-style publication, but key transitions and details felt lacking in several places, such that it was difficult to figure out how the different parts combine.

From what I can tell, the authors first discover a novel behavioral relationship between accuracy and speed in a relatively demanding task, as well as a brain-behavior relationship in rsFC. Then, being interested in the mechanism underlying this apparent relationship, they fit individualized whole-brain model to recapitulate the observed correlations and find that aspects of the discovered model correlates with aspects of behavior in the “owner” of that virtual brain. To target the mechanism more precisely, they then set up a reduced model with just two nodes and characterize the relevant phase space, finally “plugging back” the two nodes into the whole-brain model and confirm that the brain-behavior relationships are recapitulated, thereby providing evidence that the identified E/I mechanism is important. The KEY logical maneuver here, given there are no subsequent validation of the physiological mechanism discovered in silico, is that all claims of mechanistic discovery hinges on the fact that 1) the simulated brain activity matches the observed rsFC (which was explicitly optimized for), and 2) empirical group average task performance correlates with model DM performance and time (for which we are given very little details).

Given the above, I think there are some things that would drastically improve readability:

- an overview figure that lays out the logic and overall methodology of the study would be tremendously helpful. I quite liked the current supplemental fig. 2, and I think it should NOT be in the supplements, but prominently displayed, perhaps with even more details, as Fig. 1A.
- a concise paragraph summarizing the concrete contributions offered in this paper, which includes novel behavioral findings, a computational framework for fitting personalized brain models, and identified (in silico) biological mechanisms with respect to balance of long-range E & I inputs (and perhaps others). As it stands, it’s difficult to identify the specific contributions and how they relate to each other, which makes it hard to evaluate both the technical soundness, as well as the significance in the context of current literature, of said contributions. Such a paragraph is typically the last paragraph of the intro or the first paragraph of the discussion, but is currently missing due to the current formatting style the authors use.
- on the other hand, while the abstract contains a brief summary of the work, I personally found the overly ambitious and general interpretations detracted from the actual concrete results (which are interesting in their own right): have the authors really mapped “human intelligence to machines”? Given the rather weak correlation between FC and behavior in the real data, is it really true that “model simulations revealed a mechanistic link between EI balance, fluid intelligence, DM, WM, RT, and brain synchrony”? Did the model really exhibit “fluid intelligence”? Similarly, L300 in discussions: “the

research yields multiscale biophysical virtual brains that are able to perform cognitive tasks” is quite a broad statement that I do not believe to be true in generality, and is still questionable even in the limited context of the study’s results, given the lack of details on how the specific cognitive tasks are performed by the model.

- there is quite literally zero acknowledgement of limitations in the discussion section, or anywhere else: how seriously should a reader take the findings given that it is not acknowledged that the personalized brain network models are 379 coupled nodes of Wilson-Cowan style neural mass/field models? This is just one of several strong limitations that should contextualize all the mechanistic findings in the paper, especially when related to behavior, and is entirely missing. I raise more in the next section regarding concerns over technical details. Acknowledgements of such limitations are, in my view, very important, especially for a fairly general journal where readers are unlikely to be already familiar with the pros and cons of such whole-brain modeling frameworks, of both there are many.
- some important sections of the text don’t have any main figures associated with it. For example, the section “simulated brain activity correlates with...” is essential for the logic of the argument the authors want to make, but the reader can only find supplemental figures for this information. Why? Same thing for “E/I-ratios switch between stable and flexible WM”. This seems to be a central result, but is not depicted as a main figure. This created confusion for me, as the overall conclusion relies on these intermediate steps, but their importance is downgraded by a lack of dedicated main figures.
- related, a ton of important details appear to be buried in the supplementary information, which is a mixture of description of results and methods. Why? This is make is really difficult to read the paper, as “supplementary” should not contain information CRITICAL to understanding the study, such as the frontal-parietal circuit.
- the authors make multiple references to “Nobel laureate Daniel Kahneman’s” ideas regarding the two systems. I can see how this is tangentially related, but I personally found this distracting and unnecessary: it appears as an effort to appeal to authority, in this being ideas from a prominent psychologist, but the actual results of the study—conclusions drawn from model simulations that does not represent the real complexities of cognition—are quite far removed from the types of studies cited in Kahneman’s book. Are the results from this study supposed to provide support for the two-system idea, or are those ideas supposed to provide credibility to the current findings? It is even more unfortunate when a significant fraction of the relevant psychological studies appear to be irreproducible (e.g., <https://replicationindex.com/2020/12/30/a-meta-scientific-perspective-on-thinking-fast-and-slow/>).
- I cannot decipher Figure 2A, especially the top panel, while the figure caption just says “reduced model with two brain regions”. Is the FC just the correlation between two nodes then? What is “noise”? Do noise and coupling strength end up being tunable parameters in the final model, or are they set to a constant value? If so, are the key findings robust to them taking different values?
- As a general note, figure captions can be much, much more descriptive.

Concerns regarding technical details:

- how robust and meaningful is the relationship between rsFC and behavior? Are the correlation significances adjusted for multiple comparison? Why is total rsFC taken to be the brain measure, instead of thousands of other resting state fMRI metrics one could have used, even ROI-specific rsFC, given the detailed investigation on the PFC-PPC circuit? The choice of mean correlation between all regions seems out of the blue, and certainly not motivated in the introduction (possibly a presentation issue). Currently, the first place “functional connectivity” shows up is L135, and it’s entirely unclear why the authors care about this specific metric.

- similarly, there are many different model parameters (even in a SINGLE node of the coupled E/I circuit) one could have made free, why was this single parameter of the ratio between long-range excitatory strength and feed-forward inhibition chosen? This is critical, as surely fixing this parameter but varying others would have provided equally adequate fits to the empirical rsFC, if not better ones, given the existing degeneracy in fitting the 2×379^2 numbers. In other words, if varying other parameters could have resulted in similarly good fits to rsFC, is the proposed mechanism (in E/I) valid?

- how realistic is the independent fitting of long-range E and I connections? On that note, I wasn’t sure whether the FF inhibition has the constraint of being spatially local, and if not, how realistic are these long-range inhibitory connections that can be independently tuned from the excitatory ones? L428-431 seem to indicate that it’s actually the weight of the EXCITATORY onto postsynaptic INHIBITORY populations, which would make the long-range connection somewhat more plausible (but still a few references justifying this would be nice, especially the fact that they can be plastic), but it is then extremely misleading that this is called feedforward inhibition everywhere in the text and only explained in a single place buried in the methods that it’s actually excitation. Related, is it really tuning E/I ratio then? Do we know what the actual effect on the inhibitory neurons is (e.g., disinhibition), such that the local effective E/I balance is affected? I do think it’s nice that the control analysis of tuning just the LRE weights was performed.

- Also, this ratio between LRE and FFI is allowed to span 4 orders of magnitude (judging by Fig 2a): was this the case in the fitted brain models? How wide is the distribution of fitted values? Is that range realistic or not? I could not find this information anywhere in the main text nor supplemental figures, nor is it addressed as a potentially unrealistic source of variation.

- a similar point can be made about the homeostatic rule: why is 4Hz the target? Do things break down with a different target rate?

- In general, how realistic are the simulated brain time series? While we are assured that the simulated rsFC matches that of the empirical time series (again, by design), real brain activity has many other characteristic behaviors, such as a particular structure in the autocorrelation / power spectrum. While it’s entirely out of scope to require the whole-brain model to produce realistic human brain time series, to what degree the model does so is important to acknowledge, given that the overarching question is “how to map human intelligence to machines”.

- there is very little detail or motivation provided for the reduced 2-node model: why PFC and PPC? Is there something special about the specific parameter values that make this model more akin to those

two areas than, say, OFC and ACC? Or is it simply a generic two-node version of the full 379-node model?

- critically, I'm actually not sure how the two-node (nor the subsequent "multiscale" 379-node model) is doing the task? Specifically, how are the "% correct decisions" and "integration time" values in Figure 3 derived? Apologies if I had missed this, but I could not find this in the method section nor in the main text, and I believe this is a central piece of information to evaluate the soundness of the authors' claims and the mechanistic conclusions drawn from the model. A description of this in words is found in the supplements (which is definitely not where it should be), but simply writing out the equations, and the inputs and outputs, would make this much easier to evaluate the soundness of the approach.

Some other minor suggestions

- Figure 1. e-j showing the non-significant correlations (with fairly large magnitude of correlation) is a bit misleading since it's only clarified in the main text. It's easy to mistake a correlation value of 0.35 as notable, even if statistically indistinguishable from the null effect. I'd recommend removing these except for g.
- Could use more informative labels for legend in many places, instead of, e.g. PMAT24_A_CR, it'd be a lot easier for the reader to see % correct
- Figure 3 could really use a circuit schematic, especially showing how the DM "task" is abstracted for the model

Richard Gao, PhD

University of Tuebingen

POINT-BY-POINT RESPONSE

Manuscript: Learning how network structure shapes decision-making for bio-inspired computing

Authors: Michael Schirner, Gustavo Deco, Petra Ritter

Review received: 2023-02-08

We thank the Reviewers for their detailed comments, which were helpful for improving the manuscript. In the following we address all points made by the Reviewers and explain how we considered these points in the revision of the manuscript.

We would like to note that due to two missing figures and one missing section we uploaded updated manuscript files via the link "Send Manuscript Correspondence" in the Manuscript Tracking System on 2022-11-26 but are unsure whether the Reviewers saw the first or the updated version of our manuscript: Fig. 4 (Sup. Fig. 2 in the update), Supplementary Fig. 9 (Sup. Fig. 11 in the update) and the section "How E/I-ratios control FC" were missing in the first uploaded manuscript version (we marked all changed items in the updated version from 2022-11-26 with color highlighting). The two figures and section are now also contained in the revised manuscript as Fig. 4, Supplementary Fig. 9 and the section "How E/I-ratios control FC" and Reviewers may have seen them before or not. We would like to apologize for any additional work this may have caused.

Reviewer #2 (Remarks to the Author):

Summary

The authors in this work have analyzed data from the Human Connectome Project regarding correlations between general and fluid intelligence and fast and slow decision-making. The analyses revealed that individuals with higher general and fluid intelligence made easy decisions faster but took more time for complex problems. The authors then developed a computational model based on mean-field approximations and the results of their analysis on the HCP data. The authors proposed a fitting algorithm for tuning the E/I ratios within their model as well as a learning rule for feedback inhibition control. The model predicts a mechanistic link between excitation-inhibition balance, fluid intelligence, decision-making, working memory, reaction times, and brain synchrony. Furthermore, the model manifests the same trade-off between speed and accuracy as the human subjects in the Human Connectome Project.

The manuscript is well-organized and well-written. The authors can improve the figures and add more information in the captions (see my comments below). The numerical results support the authors' claims, and their model correctly predicts the empirical results. The references in this work are adequate.

We thank the Reviewer for this precise summary.

Comments

1. The authors estimated correlations between at most two variables, for instance, the g-factor vs. reaction times (RT) or functional connectivity (FC) and PMAT24_A_CR. What happens when one measures the correlation between multiple variables, such as g-factor and RT vs. FC? In other words, how do all the variables reported in the text affect FC?

We performed now multiple regression to compute the coefficient of multiple correlation to estimate how all variables reported in the text (g, PMAT24_A_CR, PMAT24_A_RT CR, ProcSpeed, CardSort) affect mean FC and now mention the result in the main text:

"Multiple regression to compute the coefficient of multiple correlation between all reported behavioral variables (g, PMAT24_A_CR, PMAT24_A_RT CR, ProcSpeed, CardSort) and FC yielded $r=0.16$ ($p<0.001$), which was only slightly higher than the univariate correlation between mean FC and PMAT24_A_RT CR."

2. In figure 2c, I assume "original" means the model without tuning. Please clarify that in the text.

Furthermore, in figure 2b, what does "lower triangular" mean? The matrix in the bottom panel of 2b does not look lower triangular.

Yes, "original" means the model without tuning and the feedforward inhibition parameters set at their default values, which we now clarify in the text:

"In comparison to the original model (Fig. 3c, green curves), where E/I-ratios were left untuned at their default settings ($w_{ij}^{LRE} = 1$ and $w_{ij}^{FFI} = 0, \forall i, j \in \{1, \dots, N\}$) from Deco et al.³³, and compared to a variant where only w_{ij}^{LRE} values were tuned (Fig. 3c, red curves), tuning both w_{ij}^{LRE} and w_{ij}^{FFI} at the same time allows to smoothly set the state of synchronization between each pair of brain regions (Fig. 4), which can be used to considerably reduce the root-mean-square error between simulated and empirical FC (Fig. 3c, blue curves)."

We now also clarify the three variants in the caption:

"c Pearson correlations and root-mean-square errors between all N=650 empirical and simulated FCs for three different model variants: EI-tuning (the tuning algorithm applied on both w_{ij}^{LRE} and w_{ij}^{FFI}), E-tuning (the tuning algorithm applied only on w_{ij}^{LRE}), original (tuning of a scalar global coupling scaling factor to rescale C_{ij})."

Regarding "lower triangular": We updated the caption to clarify that we refer to the "lower triangular portion" of the matrix and not to a lower triangular matrix:

"Empirical (upper triangular portion of the matrix) versus simulated (lower triangular portion of the matrix) FC..."

3. Please add the corresponding panels of figure 3 in lines 207 - 211, where you present the results on how amplitude and synaptic inputs affect the performance.

The corresponding panels were added (the Figure is now called "Fig. 5") and the sentences now read:

"Decreasing the mean amplitude of inputs increased decision accuracy as well as integration time (Fig. 5a, b). Similarly, increasing the correlation of input noise to the two PPC populations also led to increased performance and integration time (Fig. 5c, d). Integration times followed an inverted U-shape and were at their maxima for intermediate levels of noise correlation ($r \sim 0.5$, Fig. 5d). In contrast, input correlation to the two PFC populations had no relevant effects (Fig. 5c, d)."

4. It is unclear why the authors devised their fitting algorithm instead of a conventional one to find the optimal values for W_{LRE} and W_{FFI} .

We now clarified in the Methods section "Fitting algorithm":

"While the overall parameter optimization approach followed a standard gradient descent schema, importantly, the gradients are based on the direct monotonic and smooth relationship that we identified between E/I-ratios and FC, respectively population synchronization (Fig. 4), creating a direct biologically interpretable link between brain network structure (specifically the E/I-ratios between network nodes) and the emerging brain network dynamics when simulating the model."

5. The authors should provide more details on the learning rate schedule (Was it linear, exponential, or something else?).

We now clarify the learning rate schedule:

"The tuning lasted over six stages where each stage was simulated for 10 hours of biological time. The learning rate η_{EI} was halved and the window size of simulated FC ρ^{sim} was doubled in each stage, starting with a learning rate of $\eta_{EI}=0.1$ and a window size of 150 TRs."

Minor Comments

1. The text's abbreviations in Figure 1, Tables 1 and 2, and elsewhere are confusing.

We agree that specifically the two abbreviations “PMAT24_A_CR” (“correct responses in the test PMAT24-A”, a measure of fluid intelligence--FI) versus “PMAT24_A_RTCT” (“median response time for correct responses in the test PMAT24-A”) are hard to differentiate and it is confusing that the abbreviations “FI” and “PMAT24_A_CR” both denote “fluid intelligence”.

We found ourselves faced with the dilemma that on the one hand the test has a specific meaning (fluid intelligence score) and on the other hand, if we would rename the abbreviations as they are used throughout the HCP, then this may create more confusion and papers on HCP data become less findable.

To mitigate we now tried to make the association between high-level concepts (e.g. “fluid intelligence”) and specific test results (e.g. PMAT24_A_CR) clearer by consistently mentioning both abbreviations (FI and PMAT24_A_CR) and by augmenting their introduction in the main text like this:

*“We analyzed correlations between g-factor, FI (**PMAT24_A_CR**), RT for correct responses in the FI test (**PMAT24_A_RTCT**), and PS for 1176 participants of the Human Connectome Project (HCP) Young Adult study (Table 1)²⁰. FI was measured by the number of correct responses in PMAT^{9,16}. PS was measured by the NIH Toolbox tests Dimensional Change Card Sort¹⁸ and Pattern Completion Processing Speed¹⁹ (**CardSort_Unadj** and **ProcSpeed_Unadj**). For findability we use the same abbreviations for the cognitive tests as used in the HCP (Table 2).”*

2. The captions of the figures need some improvement. The figures, in some cases, are not self-explanatory. For instance, there needs to be more information regarding the colormaps. The histograms are hard to read. The same is true for supplementary figures (for example, S3). Please ensure that the captions provide all the necessary information for the reader to understand the content of the figures without going back and forth in the main text.

Captions were extended to be more self-explanatory. Legends are now provided next to the colormaps in Fig. 5 (former Fig. 3) and Sup. Fig. 5 (former Sup. Fig. 7). The squeezed histogram in panel g of Fig. 1 has now double the space in the figure (former Fig. 1j) and several figures were enlarged for better readability.

3. In table 2, where the authors refer to PMAT24_A_RTCT, I assume that the reference number is 15 (third column).

The ditto mark was replaced by the actual reference number.

4. The section title in line 169 seems confusing.

The section title now reads: “Simulated brain activity correlates with cognitive performance”.

5. Please comment on the computational cost of your algorithms and your simulations in general. How long does it take to run one experiment?

We now added to the section “Fitting algorithm”:

*“The wall time for simulating one hour of biological activity on one standard CPU was roughly two hours, which led to a computational cost of 6 [Stages] * 20 [CPU hours per stage] = 120 [CPU hours] to tune a single model and a total cost of 78,000 CPU hours to tune all 650 models. Fitting runs were executed in parallel on high performance computers. The costs for running subsequent DM and WM experiments with the fitted and coupled multiscale models were negligible and performed on a standard laptop as only a few seconds of activity were needed to simulate one DM or WM experiment.”*

6. The authors use the same symbol for the learning rate of the fitting algorithm and the feedback inhibition rule (η). It might be obscure to some readers, so please use different symbols.

The learning rate of the fitting algorithm is now called η_{EI} and the learning rate of FIC is now called η_{FIC} .

7. Line 473 typo (\eta not eta).

The typo was corrected (and η is now named η_{EI} to distinguish it from the FIC learning rate as per another comment).

8. The Github link to the source code is just a placeholder. Please add the source code.

We make the code available to the reviewers via this link:

https://drive.google.com/drive/folders/1SF16Drqj9Tv1up8ua8_jWaWFPMzDIC67?usp=share_link

The code will be made publicly available in the GitHub repository mentioned in the Code Availability statement upon acceptance of the paper.

We would like to thank the Reviewer again for taking the time to provide detailed comments, which helped considerably to improve the manuscript.

Reviewer #3 (Remarks to the Author):

In this paper, Schirner and colleagues use whole-brain modeling fit to individual subjects' fMRI data to investigate the mechanistic basis of fluid intelligence during complex cognitive tasks. Their main results show that:

- there exists non-trivial correlations between reaction times and general measures of intelligence (g-factor): negative correlation for easier problems (higher intelligence → faster problem solving), and positive correlation for harder problems; while there are varying degrees of correlation between resting state functional connectivity (rsFC) and task measures / intelligence, most notably, task reaction time*
- using subject-specific rsFC as the target, DTI-derived structural connectivity as the connectivity matrix, and the ratio between long-range excitation (LRE) and feedforward inhibition (FFI) as free parameters, the authors can tune individualized, whole-brain, 379-node neural mass models with local E/I populations such that the simulated fMRI data exhibits similar rsFC patterns for all subjects (in terms of correlation and RMSE)*
- there are correlations between task measures (at the individual and stratified levels), such as RT, and the simulated dynamics of the individualized brain models, in particular mean synaptic amplitude and correlation*
- in a 2-node model of PFC-PPC interaction, the authors identify synaptic input amplitude and coupling degree (correlation) as important parameters for task performance and speed, in a nonlinear fashion; they then insert the 2-node model into the whole-brain model such that the simulated synaptic currents now drive the 2-node model, and find that simulated performance and speed match empirical data at the stratified level*
- they further validate the computational findings by refitting the model many times, observing similar simulated time series and directions of correlation despite apparent variance in fitted parameter values*

I think this is an interesting paper that aims to combine neurophysiological mechanisms, computational modeling of brain dynamics and functional connectivity, and cognition. If I'm not mistaken, the main contributions can be summarized as "personalized virtual brain model reproduces subject-specific rsFC patterns, as well as nonlinear relationship between speed and performance in fluid intelligence tasks (as well as the relationship between neural and behavior data), and suggests aspects of synaptic dynamics, driven by the balance between long-range E and I strengths, as important contributing factors." In my opinion, this type of work that bridges neural mechanisms and complex cognition is valuable for neuroscience, even if obvious abstractions on the modeling side are necessary. Additionally, I particularly appreciated the "model validation" section, as ensuring the robustness of results in a computational work is crucial.

We thank the Reviewer for this summary and overall positive evaluation.

On the other hand, I have numerous major concerns regarding the framing of the results, in particular the mismatch between the claimed significance and the actual results (and a lack of acknowledgement there), and the many ambiguities surrounding crucial technical details (expanded below). These issues are compounded by what I personally found to be a difficult presentation, both in terms of the organization of the narrative at a high level, as well as a lack of key main figures and descriptive figure captions. I apologize if I had misinterpreted any of the results in my summary, I simply could not find a lot of information in the main text and methods, but if the above are accurate in the authors' eyes, then I think the paper could be improved on the presentation front in order to make it easier for a reader to digest the main messages. As a whole, I found the paper to have some interesting results and refreshing approaches, but is extremely over-interpreted, lacking in crucial methodological details, and presented in a very confusing way. I outline my major concerns below, from which I hope the authors can derive some helpful suggestions.

We thank the Reviewer for raising these points. We strived to improve the framing of the results and the presentation of technical details and answer below where details on each item are outlined.

Concerns regarding presentation and framing:

I appreciate that the authors are undertaking a difficult task in framing a paper which encapsulates results across neuroimaging, computational modeling, and behavioral analysis. Nevertheless, the paper currently reads as if it was written for a condensed Nature or Science-style publication, but key

transitions and details felt lacking in several places, such that it was difficult to figure out how the different parts combine.

We agree and can see how the brevity of the main text (indeed written for a condensed Nature-style publication) made it difficult to see how the different parts combine and are grateful for the Reviewer to point this out. We accordingly restructured the manuscript and pulled Supplementary section “How E/I-ratios control FC” and the associated figure (now Fig. 4) as well as the overview/schematic figure (now Fig. 2) into the main text to make full use of the given space limits, providing now more details directly in the main text. As per another comment below we also added a concise paragraph summarizing the concrete contributions of the manuscript and the order in which they appear as the last paragraph of the Introduction to make the structure clearer from the beginning. We generally augmented figure captions and added transitory text and explanatory comments at various places to improve presentation. We now more explicitly refer to variables and equations and other methodological details to make technical details clearer.

From what I can tell, the authors first discover a novel behavioral relationship between accuracy and speed in a relatively demanding task, as well as a brain-behavior relationship in rsFC. Then, being interested in the mechanism underlying this apparent relationship, they fit individualized whole-brain model to recapitulate the observed correlations and find that aspects of the discovered model correlates with aspects of behavior in the “owner” of that virtual brain. To target the mechanism more precisely, they then set up a reduced model with just two nodes and characterize the relevant phase space, finally “plugging back” the two nodes into the whole-brain model and confirm that the brain-behavior relationships are recapitulated, thereby providing evidence that the identified E/I mechanism is important. The KEY logical maneuver here, given there are no subsequent validation of the physiological mechanism discovered in silico, is that all claims of mechanistic discovery hinges on the fact that 1) the simulated brain activity matches the observed rsFC (which was explicitly optimized for), and 2) empirical group average task performance correlates with model DM performance and time (for which we are given very little details).

We would like to point out that the “two-node model” is different than the DM circuit. The two-node model is a simplified version of the 379-nodes large-scale network with just two nodes to create tuning curves for different E/I-ratios. In contrast, the DM circuit is an established model for winner-take-all DM and persistent firing WM (e.g. Wang XJ, 2002, Neuron; Wong KF, Wang XJ, 2006, JNeuro; Murray et al., 2017, JNeuro).

The mechanism of E/I-balance and its effect on FC, synchronization, and amplitude of population activity is targeted with the reduced two-node model to produce the tuning curves in Fig. 4. This was done to estimate the effect of tuning E/I-balance in isolation using a simpler network than the 379-nodes network, to disentangle the effect of more than two nodes. In contrast, the winner-take-all competition mediated by cross-inhibition mechanism for decision-making and the persistent-firing via recurrent excitation mechanism of WM was targeted with the DM circuit, which was likewise inspired and validated by empirical data in previous works.

What is “plugged” into the 379-nodes large-scale model is the DM circuit, not the reduced two-node model.

We now better clarify the distinction between the two networks in the main text:

“The two-node model is a simplified version of the 379-node large-scale brain model to study the effect of large-scale E/I-balance with a simpler network structure (Fig. 4c-j). The two-node model is unrelated to the functional frontoparietal decision-making circuit³⁷ (DM circuit) further introduced below. The two-node model simulated mutual and recurrent interaction between one excitatory and one inhibitory population as in the 379-nodes large-scale model, but with a simpler network of only two nodes to produce tuning curves (Fig. 4c-h). In contrast, the DM circuit is an existing frontoparietal circuit model to simulate winner-take-all competition resulting from cross-inhibition of two excitatory populations via one inhibitory population, which we studied in isolation (Fig. 5), and after coupling with the 379-nodes large-scale model to form the multiscale model (Fig. 6).”

We would like to point out that the physiological mechanism discovered in silico was extensively analyzed with phase space plots in Supplementary Figs. 4, 5, and 6 and animated phase spaces in Supplementary Movie 1 and described in main text and Supplementary sections. The observation that the large-scale model as well as the DM circuit generate valid predictions of empirical observations (Figs. 3, 4, 5, 6) was for us reason to extract and describe the model mechanism as a candidate mechanism. We now suggest in the Discussion as a next step the extracted mechanisms may be further validated or falsified with empirical data:

“It must be mentioned that BNMs are high-dimensional models with thousands of parameters and the identified mechanism may be one out of a potentially infinite number of mechanisms that could explain the observed data. As with any scientific hypothesis, it is therefore crucial to validate and falsify theory with dedicated experiments.

Since the used brain network model simulates detailed properties of neural systems like input currents, firing rates, synaptic activity, and fMRI, it is directly amenable for further validation or falsification with empirical data from different modalities.”

Given the above, I think there are some things that would drastically improve readability:

- an overview figure that lays out the logic and overall methodology of the study would be tremendously helpful. I quite liked the current supplemental fig. 2, and I think it should NOT be in the supplements, but prominently displayed, perhaps with even more details, as Fig. 1A.*

We now display the former Supplementary Fig. 2 prominently as Fig. 2 (“Modelling outline”) to lay out the logic and overall methodology and added more details to the caption.

- a concise paragraph summarizing the concrete contributions offered in this paper, which includes novel behavioral findings, a computational framework for fitting personalized brain models, and identified (in silico) biological mechanisms with respect to balance of long-range E & I inputs (and perhaps others). As it stands, it’s difficult to identify the specific contributions and how they relate to each other, which makes it hard to evaluate both the technical soundness, as well as the significance in the context of current literature, of said contributions. Such a paragraph is typically the last paragraph of the intro or the first paragraph of the discussion, but is currently missing due to the current formatting style the authors use.*

We now added a concise paragraph summarizing the concrete contributions of the manuscript as the last paragraph of the Introduction:

“In the following, we first provide behavioral findings that link intelligence test results with processing speed and FC (Fig. 1, Table 1). Then we demonstrate a computational framework for closely fitting BNMs to personal FC (Fig. 2, 3), and subsequently explain the empirical data based on the in silico identified biological candidate mechanisms (Figs. 4-6 and Sup. Figures). For the fitting we created a parameter learning algorithm that makes use of our observation that FC and synchronization between two simulated brain areas can be smoothly and monotonically tuned via their long-range excitation-inhibition balance (E/I-ratio). We then show that the internal dynamics of the fitted models correlated with the empirical cognitive performance of the subjects (Fig. 4a, b). In addition, E/I-balance modulated the amplitude and synchrony of large-scale synaptic currents in a way that modulated DM winner-take-all races and WM persistent activity in accordance with the empirical observations (Figs. 5, 6 and Sup. Fig. 4). Phase space analysis of the resulting model dynamics allowed to frame the trade-off between speed and accuracy in terms of generic dynamical systems behavior in dependence of the E/I-balance of long-range brain network topology, which may jointly explain individual variability in FC, intelligence, and processing speed (Supplementary Figs. 5, 6 and Supplementary Movie 1).”

- on the other hand, while the abstract contains a brief summary of the work, I personally found the overly ambitious and general interpretations detracted from the actual concrete results (which are interesting in their own right): have the authors really mapped “human intelligence to machines”?*

We removed the question that we asked in the Abstract, “How can we map intelligence to machines?”, and instead now write:

“To better understand how network structure shapes intelligent behavior, we developed a learning algorithm...”

Given the rather weak correlation between FC and behavior in the real data, is it really true that “model simulations revealed a mechanistic link between EI balance, fluid intelligence, DM, WM, RT, and brain synchrony”?

The identified mechanistic link exists independent of any correlations with empirical data as it describes a chain of events that is explicitly implemented as a computational model that can be reproduced and demonstrated, capturing the entire logic of the theory.

We briefly recapitulate the main links of the mechanism here:

- **E/I-balance** modulates **brain synchrony** / FC, and amplitude of synaptic currents
 - Higher E/I ratio leads to higher synchrony / lower amplitude
- Lower synaptic current amplitude, respectively, higher current synchrony, lead to **slower RT** and **higher DM percent correct decisions in winner-take-all races**
 - Decreased synaptic current amplitudes create a new low-activity attractor in phase space that allows to prolong evidence integration for decision-making (vice versa, increased inputs create a new high-activity saddle that quickly activates both decision populations)
 - Higher correlation leads to more diagonal flow in phase space, keeping decision options more open for further evidence integration

Regarding correlations with empirical data, we would argue that several of the found correlations in Table 1 and Fig. 1 are quite large compared to what is typically found in intelligence studies (e.g. Vieira et al. cited below reviewing that “studies predicting fluid intelligence averaged $r = 0.15$ ”):

1. Sheppard, Leah D., and Philip A. Vernon. "Intelligence and speed of information-processing: A review of 50 years of research." *Personality and individual differences* 44.3 (2008): 535-551.
2. Vieira, B. H., Pamplona, G. S. P., Fachinello, K., Silva, A. K., Foss, M. P., & Salmon, C. E. G. (2022). On the prediction of human intelligence from neuroimaging: A systematic review of methods and reporting. *Intelligence*, 93, 101654.
3. Deary, Ian J., Geoff Der, and Graeme Ford. "Reaction times and intelligence differences: A population-based cohort study." *Intelligence* 29.5 (2001): 389-399.
4. Ferguson, Michael A., Jeffrey S. Anderson, and R. Nathan Spreng. "Fluid and flexible minds: Intelligence reflects synchrony in the brain's intrinsic network architecture." *Network Neuroscience* 1.2 (2017): 192-207.
5. Popp, J. L., Thiele, J. A., Faskowitz, J., Seguin, C., Sporns, O., & Hilger, K. (2023). Structural-Functional Brain Network Coupling Predicts Human Cognitive Ability. *bioRxiv*, 2023-02.
6. Omidvarnia, A., Sasse, L., Larabi, D. I., Raimondo, F., Hoffstaedter, F., Kasper, J., ... & Patil, K. R. (2023). Is resting state fMRI better than individual characteristics at predicting cognition?. *bioRxiv*, 2023-02.

Did the model really exhibit “fluid intelligence”?

We believe we never claimed that the “model exhibits fluid intelligence”. We believe that we found a relationship between brain network architecture, excitation-inhibition balance and fluid intelligence and can demonstrate this link with the implemented model and learning algorithm. We would be glad if the Reviewer could point us to specific paragraphs where such a claim seems to be made and we would be happy to revise accordingly.

Similarly, L300 in discussions: “the research yields multiscale biophysical virtual brains that are able to perform cognitive tasks” is quite a broad statement that I do not believe to be true in generality, and is still questionable even in the limited context of the study’s results, given the lack of details on how the specific cognitive tasks are performed by the model.

We reformulated the statement:

“The research yields an implementation of multiscale brain network models that are able to perform decision-making tasks, both of which have recently been identified as crucial steps to explain the relationship between microscopic phenomena, large-scale brain function, and behavior as well as generating brain digital twins for personalized medical interventions47.”

• there is quite literally zero acknowledgement of limitations in the discussion section, or anywhere else: how seriously should a reader take the findings given that it is not acknowledged that the personalized brain network models are 379 coupled nodes of Wilson-Cowan style neural mass/field models? This is just one of several strong limitations that should contextualize all the mechanistic findings in the paper, especially when related to behavior, and is entirely missing. I raise more in the next section regarding concerns over technical details. Acknowledgements of such limitations are, in my view, very important, especially for a fairly general journal where readers are unlikely to be already familiar with the pros and cons of such whole-brain modeling frameworks, of both there are many.

We added a new paragraph to the discussion section to discuss limitations:

"It must be mentioned as a limitation that BNMs are high-dimensional models with thousands of parameters and the identified mechanism may be one out of a potentially infinite number of mechanisms that could explain the observed data. As with any scientific hypothesis, it is therefore crucial to validate and falsify theory with dedicated experiments. Since the used brain network model simulates detailed properties of neural systems like input currents, firing rates, synaptic activity, and fMRI, it is directly amenable for further validation or falsification with empirical data from different modalities. By integrating diverse empirical findings into a unifying computational framework that can be iteratively refined (or refuted) dynamic models provide an avenue out of the 'reproducibility crisis'. BNMs are limited when it comes to their resolution, as they are typically based on connectivity data obtained from non-invasive imaging techniques like MRI and limited computational power to simulate large networks. These problems are addressed with multiscale models where only some parts of the brain are simulated at a finer scale (for example, at the level of spiking neurons⁴⁹) while the remaining parts are simulated by a coarser network to save computational resources. In addition, by integrating connectivity and other microstructural information from finer scale studies, for example, from invasive rodent studies⁵⁰ or post-mortem human atlases⁵¹, it becomes possible to further constrain parameters and test the plausibility of simulation results. In this regard, we note that the described relationship between E/I-balance and FC (respectively population synchronization) appears independent of the spatial and temporal scales of the network, and may be used to generally tune also finer-scale or coarser-scale networks, as it is based on generic dynamical primitives of neural mass action applicable to describe dynamics across spatial and temporal scales⁵². Although BNMs employ abstractions, like all models, further advances may emerge precisely where the assumptions break down. For example, the used ensemble models capture neural population dynamics primarily when coherence is sufficiently weak that individual spikes can be ignored or when coherence is sufficiently strong that variance can be considered small, while scale-free dynamics with unbounded variance resist mean-field reductions and may require alternative ensemble methods^{7,8}. Despite these limitations, BNMs are in contrast with artificial neural networks specifically designed to explain the underlying biology, using typically observed features of the empirical system as targets for validation and falsification (Supplementary Fig. 10) to achieve an incrementally improved computer model of the empirical system."

We additionally would like to point out that 379 coupled neural mass models is a relatively typical number compared to state of the art connectome analysis and brain modelling, where the number usually ranges between 60 and 1000 (please see references below). Especially, if we compare them with Dynamic Causal Models which usually only have a handful of regions. The number is on an order of magnitude that is typical not only for theoretical modelling studies but importantly also for purely empirical connectomics studies. We agree nevertheless that it would be helpful if brain models become much more detailed in the future. In this regard we are happy to report that the fitting algorithm seems to be independent of the network size: we were able to use it successfully in much larger networks (unpublished tests). Therefore, we expect that in the future it may be used in more detailed brain models. The reason why we opted for 379 regions is that this is based on the Human Connectome Project / Glasser et al. parcellation, which is a widely used atlas for parcellating full brains. Importantly, the Glasser atlas is based on a multimodal data classification which allows to better identify and map functional regions across subjects, which is helpful to define functional network nodes for the model.

References on typical network sizes / connectome parcellations:

- Sporns, Olaf, Giulio Tononi, and Rolf Kötter. "The human connectome: a structural description of the human brain." *PLoS computational biology* 1.4 (2005): e42.
- Glasser, M. F., Coalson, T. S., Robinson, E. C., Hacker, C. D., Harwell, J., Yacoub, E., ... & Van Essen, D. C. (2016). A multi-modal parcellation of human cerebral cortex. *Nature*, 536(7615), 171-178.
- Bullmore, Ed, and Olaf Sporns. "Complex brain networks: graph theoretical analysis of structural and functional systems." *Nature reviews neuroscience* 10.3 (2009): 186-198.
- Fornito, A., Zalesky, A., & Breakspear, M. (2013). Graph analysis of the human connectome: promise, progress, and pitfalls. *Neuroimage*, 80, 426-444.
- Hagmann, P., Cammoun, L., Gigandet, X., Meuli, R., Honey, C. J., Wedeen, V. J., & Sporns, O. (2008). Mapping the structural core of human cerebral cortex. *PLoS biology*, 6(7), e159.
- Honey, C. J., Sporns, O., Cammoun, L., Gigandet, X., Thiran, J. P., Meuli, R., & Hagmann, P. (2009). Predicting human resting-state functional connectivity from structural connectivity. *Proceedings of the National Academy of Sciences*, 106(6), 2035-2040.
- Sporns, Olaf, Christopher J. Honey, and Rolf Kötter. "Identification and classification of hubs in brain networks." *PloS one* 2.10 (2007): e1049.

• some important sections of the text don't have any main figures associated with it. For example, the section "simulated brain activity correlates with..." is essential for the logic of the argument the authors

want to make, but the reader can only find supplemental figures for this information. Why? Same thing for “E/I-ratios switch between stable and flexible WM”. This seems to be a central result, but is not depicted as a main figure. This created confusion for me, as the overall conclusion relies on these intermediate steps, but their importance is downgraded by a lack of dedicated main figures.

The main figure panels corresponding to the section “Simulated brain activity correlates with...” were now added as main Fig. 4a, b. In addition, the section “How E/I-ratios control FC” and the associated figure panels (now Fig. 4c-j) were now moved from the Supplementary to the main text, now reaching the limits for main text word count. We didn’t pull Supplementary Fig. 6 to the main text as we would like to put a higher focus on DM and the accompanying phase space analyses (Supplementary Figs. 4 and 5).

• related, a ton of important details appear to be buried in the supplementary information, which is a mixture of description of results and methods. Why? This is make is really difficult to read the paper, as “supplementary” should not contain information CRITICAL to understanding the study, such as the frontal-parietal circuit.

The Supplementary Information contains additional information that were not part of the main text due to space limitations (the main text should be no more than 6,000 words). The Supplementary sections are all referenced from the main text and organized in a way to provide supporting information to the main results. Especially the mentioned frontoparietal circuit (DM circuit), as it is an established result in the literature (please see some references above), and all details can be found in the existing publications, we did not add extensive descriptions to the main text and only provided some general information as a courtesy to the reader, as we acknowledge that the work spans a wide range of concepts from computational neuroscience.

• the authors make multiple references to “Nobel laureate Daniel Kahneman’s” ideas regarding the two systems. I can see how this is tangentially related, but I personally found this distracting and unnecessary: it appears as an effort to appeal to authority, in this being ideas from a prominent psychologist, but the actual results of the study—conclusions drawn from model simulations that does not represent the real complexities of cognition—are quite far removed from the types of studies cited in Kahneman’s book. Are the results from this study supposed to provide support for the two-system idea, or are those ideas supposed to provide credibility to the current findings? It is even more unfortunate when a significant fraction of the relevant psychological studies appear to be irreproducible (e.g., <https://replicationindex.com/2020/12/30/a-meta-scientific-perspective-on-thinking-fast-and-slow/>).

Our idea was to embed these findings into a larger context to provide an interesting background for the reader and to provide them with related and widely known ideas. We believe that readers are interested in connecting results with the historical context, which we provide non only via reference to Kahneman, but also for example via reference to other prominent and widely cited results from the field of psychology and others (e.g., References 1-6, 10-15) – which we do in order to demonstrate the impact and the long history (dating back over 130 years) of associated research streams. We checked the referenced blog but didn’t find indication that the established findings around intelligence and processing speed and the trade-off between speed and accuracy in psychology and other fields, were refuted or irreproducible. In addition, we ourselves reproduced the crucial result again with the empirical HCP data, as shown in the manuscript (Fig. 1, Table 1).

To better exclude that we appear like we would try to appeal to authority we now explicitly removed the mentioning of the name “Kahneman” from the article.

1. The sentence

“Simulation results indicate fast and slow modes of information processing that closely resemble the two systems from Nobel laureate Daniel Kahneman’s influential theory on fast and slow thinking 7.”

now reads

“Simulation results indicate that decision-making speed is traded with accuracy, resembling influential theories from the fields of economy and psychology on fast and slow thinking9.”

2. We removed the sentences

“A related idea formulated in Nobel laureate Daniel Kahneman’s influential work on “Thinking, fast and slow” contrasts two mental systems that also differ with regard to speed and depth. System 1 operates automatically and quickly with little or no effort and no sense of voluntary control. System 2, on the other hand, allocates attention to the effortful mental activities that demand it, which is often associated with the subjective experience of agency and choice. According to Kahneman mistakes occur if we allow System 1 to process a cognitive decision while we should have really passed this decision over to our slow System 2.”

and instead now write

“The idea that decision-making speed is traded with accuracy is supported by numerous empirical findings in the fields of economy, ecology, psychology, and neuroscience53–56.”

- *I cannot decipher Figure 2A, especially the top panel, while the figure caption just says “reduced model with two brain regions”. Is the FC just the correlation between two nodes then? What is “noise”? Do noise and coupling strength end up being tunable parameters in the final model, or are they set to a constant value? If so, are the key findings robust to them taking different values?*

We now better clarify definitions of FC, noise, coupling strength, whether they are tunable, and whether their tuning would impact the result in the caption:

“Fig. 3 | Identification of a smooth, monotonic relationship between E/I-ratio and FC to fit brain network models. a Tuning curves for a reduced model with only two nodes, but otherwise identical to the 379-nodes BNM. FC (that is, correlation) between the two nodes increased smoothly and monotonically as a function of their E/I-ratio $\frac{w_{1,2}^{LRE}}{w_{1,2}^{FFI}}$. The relationship between E/I-ratio and FC persisted when the strength of noise σ (upper panel; equations 5, 6) and the strength of structural coupling C_{ij} (lower panel; equations 1, 2) were modulated for test purposes (both are fixed parameters during the fitting of the full 379-nodes model). **b** Fitting results for the full 379-nodes model for one exemplary FC. Empirical (upper triangular portion of the matrix) versus simulated (lower triangular portion of the matrix) FC and joint distributions without E/I-tuning (upper panel) and with E/I-tuning (lower panel). **c** Pearson correlations and root-mean-square errors between all $N=650$ empirical and simulated FCs for three different model variants: E/I-tuning (the tuning algorithm applied on both w_{ij}^{LRE} and w_{ij}^{FFI}), E-tuning (the tuning algorithm applied only on w_{ij}^{LRE}), original (tuning of a scalar global coupling scaling factor to rescale C_{ij}).”

- *As a general note, figure captions can be much, much more descriptive.*

Following up on this and related comments by Reviewer #2, we reworked and expanded several captions. All changes are highlighted with color in the revised manuscript.

Concerns regarding technical details:

- *how robust and meaningful is the relationship between rsFC and behavior?*

The correlation coefficients between rsFC and behavior can be robustly recomputed as they do not depend on stochastic elements. The “meaning” of the relationship we tried to elucidate with the presented computational model and extracted mechanism that show how E/I-ratio can simultaneously modulate both rsFC as well as decision-making time and performance.

Are the correlation significances adjusted for multiple comparison?

We now corrected for multiple comparisons and updated the captions: “ $p < 0.05$, $p < 0.01$, $p < 0.001$; including only significant p -values (adjusted using the Benjamini-Hochberg procedure with a False Discovery Rate of 0.1).”

Why is total rsFC taken to be the brain measure, instead of thousands of other resting state fMRI metrics one could have used, even ROI-specific rsFC, given the detailed investigation on the PFC-PPC circuit? The choice of mean correlation between all regions seems out of the blue, and certainly not motivated in the introduction (possibly a presentation issue). Currently, the first place “functional connectivity” shows up is L135, and it’s entirely unclear why the authors care about this specific metric.

The choice for mean rsFC was directly based on the mechanism itself: we found that functional connectivity directly depends on the E/I balance of neural networks. We think this is important because the usage of functional connectivity as a tool for neuroscience is often criticized (e.g., FC as an epi phenomenon) and its relation to structural brain organization is often described as unclear. Here we can show a direct mechanistic dependence of FC vis-à-vis neural network architecture: it’s possible to smoothly and precisely tune FC from low to high -- by setting the E/I balance of the connection. The choice for mean rsFC therefore directly derives from the modelling results: for a two-node model the FC is just a single value to characterize synchronization in the entire system. For the 379-nodes model the average FC is a comparable compact univariate metric to characterize the entire FC matrix. Similar to the single value we get for the 2-node model, it allows a compact characterization of the average state of synchronization between all nodes. In the future we would like to deepen our investigations towards other resting-state fMRI metrics and ROI-specific measures.

To better clarify these aspects in the manuscript, we added the following paragraph to the discussion:

“Summarizing, in the present work we identified a monotonic and smooth relationship between the structural and the functional architecture of neural networks: by tuning the E/I-ratio it became possible to precisely and simultaneously tune the FC between any pair of network nodes to the desired target configuration from full antisynchronization to full synchronization. We believe this is important, as the link between FC and structural brain architecture is often described as unclear and many research streams aim for inferring structural network topology^{75,76}. We therefore expect that the described smooth and monotonic link between network architecture and FC, and the derived learning rule, will be useful to better understand and infer structural network mechanisms underlying healthy and pathological cognition^{77,78}.”

In addition, we now introduce FC in the Introduction:

“Importantly, by identifying a smooth and monotonous relationship between structural and functional neural network architecture it was possible to devise a network fitting algorithm that allows to simultaneously and precisely control the state of synchronization between every pair of network nodes, allowing to tune each connection from full antisynchronization to full synchronization, enabling a close reproduction of whole-brain subject-specific FC.”

In addition, we now motivate the choice of total FC:

“We have selected mean FC for the subsequent analyses as it is a compact representation of whole-brain FC and related to E/I-balance per our analysis (Figs. 3, 4).”

• similarly, there are many different model parameters (even in a SINGLE node of the coupled E/I circuit) one could have made free, why was this single parameter of the ratio between long-range excitatory strength and feed-forward inhibition chosen? This is critical, as surely fixing this parameter but varying others would have provided equally adequate fits to the empirical rsFC, if not better ones, given the existing degeneracy in fitting the 2×379^2 numbers. In other words, if varying other parameters could have resulted in similarly good fits to rsFC, is the proposed mechanism (in E/I) valid?

The E/I-ratio was chosen because a direct, smooth, and monotonous relationship between E/I-ratio and FC was identified that appears biologically meaningful and that was used to tune the state of synchronization between every network node from fully antisynchronized to fully synchronized. We found this link between the architecture of neural networks and the emerging dynamics and functional networks remarkable as we are not aware that it existed before in the literature, and therefore deserving of further investigation. Another reason for investigating

E/I-ratios is that they are a widely analysed metric in neurosciences with many identified links to cognition (the Discussion collects some references).

We don't see how the validity of the proposed mechanism would depend on whether other parameters in the model can be varied or not. Yes, it's possible that the same fit can be achieved by other mechanisms, there could be an infinite number of different mechanisms, in line with the Universal Approximation Theorem for neural networks, but the fact that other mechanisms can exist doesn't automatically invalidate the particular mechanism we are proposing in the manuscript. An explanation is not automatically invalid because there may exist alternative explanations. As with any research output, the mechanism we propose is a hypothesis that needs to be further validated or falsified with empirical data and/or by deriving further theoretical implications and comparing them with existing data.

To better highlight this point, we added the following sentence to the Discussion:

"It must be mentioned as a limitation that BNMs are high-dimensional models with thousands of parameters and the identified mechanism may be one out of a potentially infinite number of mechanisms that could explain the observed data. As with any scientific hypothesis, it is therefore crucial to validate and falsify theory with dedicated experiments. Since the used brain network model simulates detailed properties of neural systems like input currents, firing rates, synaptic activity, and fMRI, it is directly amenable for further validation or falsification with empirical data from different modalities. By integrating diverse empirical findings into a unifying computational framework that can be iteratively refined (or refuted) dynamic models provide an avenue out of the 'reproducibility crisis'."

• how realistic is the independent fitting of long-range E and I connections? On that note, I wasn't sure whether the FF inhibition has the constraint of being spatially local, and if not, how realistic are these long-range inhibitory connections that can be independently tuned from the excitatory ones? L428-431 seem to indicate that it's actually the weight of the EXCITATORY onto postsynaptic INHIBITORY populations, which would make the long-range connection somewhat more plausible (but still a few references justifying this would be nice, especially the fact that they can be plastic), but it is then extremely misleading that this is called feedforward inhibition everywhere in the text and only explained in a single place buried in the methods that it's actually excitation. Related, is it really tuning E/I ratio then? Do we know what the actual effect on the inhibitory neurons is (e.g., disinhibition), such that the local effective E/I balance is affected? I do think it's nice that the control analysis of tuning just the LRE weights was performed.

Connections are not fitted independently, and long-range connections are not inhibitory.

"Feedforward inhibition" is a common term to indicate that inhibitory interneurons and their target cells are activated by the same excitatory input, e.g.

- Buzsáki, György. "Feed-forward inhibition in the hippocampal formation." *Progress in neurobiology* 22.2 (1984): 131-153.
- Pouille, Frédéric, and Massimo Scanziani. "Enforcement of temporal fidelity in pyramidal cells by somatic feed-forward inhibition." *Science* 293.5532 (2001): 1159-1163.
- Ferrante, Michele, Michele Migliore, and Giorgio A. Ascoli. "Feed-forward inhibition as a buffer of the neuronal input-output relation." *Proceedings of the National Academy of Sciences* 106.42 (2009): 18004-18009.

The concept of feedforward inhibition was already defined in the original manuscript, not only in the Methods, but also early on in the Results section, at the place where the term was first introduced

("Importantly, we added feedforward inhibition to increase biological realism 21–31: while in previous BNM studies there was typically only long-range coupling between excitatory populations, here, excitatory masses additionally targeted inhibitory populations"; in addition, see the visualization in former Sup. Fig. 2b and corresponding caption "In previous BNM studies long-range white matter coupling from excitatory to inhibitory populations was often absent. By adding these connections, it became possible to precisely tune the E/I-ratio of synaptic inputs between each pair of BNM nodes").

Nevertheless, we now made the definition more explicit in the subsequent sentences in the Results section:

"By tuning the ratio of long-range excitation (LRE; strength of long-range excitatory-to-excitatory coupling w_{ij}^{LRE} , equation 1) to feedforward inhibition (FFI; strength of long-range excitatory-to-inhibitory coupling w_{ij}^{FFI} , equation 2) ..."

We provided 10 references already in the original manuscript (and kept them in the revision) justifying the existence of long-range excitatory to postsynaptic inhibitory connections:

“Importantly, we added feedforward inhibition to increase biological realism^{24–34}: while in previous BNM studies there was typically only long-range coupling between excitatory populations, here, excitatory masses additionally targeted inhibitory populations (Fig. 2b and Methods).

...

24. Yang, W., Carrasquillo, Y., Hooks, B. M., Nerbonne, J. M. & Burkhalter, A. Distinct balance of excitation and inhibition in an interareal feedforward and feedback circuit of mouse visual cortex. *J. Neurosci.* 33, 17373–17384 (2013).
25. Anastasiades, P. G. & Carter, A. G. Circuit organization of the rodent medial prefrontal cortex. *Trends Neurosci.* (2021).
26. Sermet, B. S. et al. Pathway-, layer- and cell-type-specific thalamic input to mouse barrel cortex. *Elife* 8, e52665 (2019).
27. Åhrlund-Richter, S. et al. A whole-brain atlas of monosynaptic input targeting four different cell types in the medial prefrontal cortex of the mouse. *Nat. Neurosci.* 22, 657–668 (2019).
28. Lee, S., Kruglikov, I., Huang, Z. J., Fishell, G. & Rudy, B. A disinhibitory circuit mediates motor integration in the somatosensory cortex. *Nat. Neurosci.* 16, 1662–1670 (2013).
29. Naskar, S., Qi, J., Pereira, F., Gerfen, C. R. & Lee, S. Cell-type-specific recruitment of GABAergic interneurons in the primary somatosensory cortex by long-range inputs. *Cell Rep.* 34, 108774 (2021).
30. Wall, N. R. et al. Brain-wide maps of synaptic input to cortical interneurons. *J. Neurosci.* 36, 4000–4009 (2016).
31. Zhang, S. et al. Long-range and local circuits for top-down modulation of visual cortex processing. *Science* 345, 660–665 (2014).
32. Cruikshank, S. J., Urabe, H., Nurmikko, A. V & Connors, B. W. Pathway-specific feedforward circuits between thalamus and neocortex revealed by selective optical stimulation of axons. *Neuron* 65, 230–245 (2010).
33. Frandolig, J. E. et al. The synaptic organization of layer 6 circuits reveals inhibition as a major output of a neocortical sublamina. *Cell Rep.* 28, 3131–3143 (2019).
34. Barbas, H. General cortical and special prefrontal connections: principles from structure to function. *Annu. Rev. Neurosci.* 38, 269–289 (2015).”

Yes, it's really tuning of E/I-ratios, this can be seen in Figure 3c, where the LRE and FFI parameter values for different E/I-ratios are displayed.

Yes, we analyzed how the effect on inhibitory neurons affects local effective E/I, and provide the results in the sections “How E/I-ratios control FC” and “How E/I-ratios control synchrony and amplitude of synaptic currents”.

• Also, this ratio between LRE and FFI is allowed to span 4 orders of magnitude (judging by Fig 2a): was this the case in the fitted brain models? How wide is the distribution of fitted values? Is that range realistic or not? I could not find this information anywhere in the main text nor supplemental figures, nor is it addressed as a potentially unrealistic source of variation.

The E/I-ratio as it was defined in the manuscript refers to the ratio of the long-range coupling parameters w_{LRE} versus w_{FFI} and not to the ratio of all excitatory versus all inhibitory synaptic currents arriving at the excitatory populations (in addition to long-range currents, there are also local currents). The ratio of the coupling parameters w_{LRE} versus w_{FFI} can span the entire range of values from zero to infinity as either one of these parameters approaches zero (in this regard, it may be interesting to note that previous brain network modelling studies typically used only LRE connections and no FFI connections at all – corresponding to an E/I - parameter-ratio that approaches infinity).

The ratios of the two parameters w_{LRE} and w_{FFI} – since one of them can become zero – can occupy a much wider range than the E/I ratios of the resulting currents themselves. In contrast, the ratio of all excitatory versus all inhibitory currents (long-range plus local) is bound to a narrow/balanced range because (i) the sigmoidal activation function that translates incoming currents into firing rates binds the resulting activation to a narrow range and (ii) there exists the local excitatory and inhibitory connectivity that adds to the total sums of excitatory and inhibitory input currents. Importantly, the local inhibitory to excitatory coupling is controlled by Feedback Inhibition Control such that there is always a balanced amount of excitation versus inhibition arriving at the excitatory populations such that excitatory populations show a long-term average firing rate of 4 Hz – independent of the settings of w_{LRE} and w_{FFI} . That is, the E/I-ratio of all incoming excitatory versus inhibitory currents (in contrast to the ratio of the parameters w_{LRE} and w_{FFI}) is always balanced due to local feedback inhibition control.

To better clarify this point in the manuscript, we added the following sentence:

“It is important to point out that ‘E/I-ratio’ here refers only to the ratio of the long-range coupling strength parameters $\frac{w_{ij}^{LRE}}{w_{ij}^{FFI}}$ without considering the effect of local inhibitory connectivity J_i . Due to FIC the E/I-ratio of the total sums of long-range and local currents that arrive at excitatory populations ($\frac{W_{E I_0 + w + J_{NMDA} S_i^E + J_{NMDA} \sum_j w_{ij}^{LRE} C_{ij} S_j^E}{J_i S_i^I}$, equation 1) is always in a balanced state, which ensures an average firing rate of 4 Hz of the excitatory population even in the case that long-range connections are unbalanced.”

- *a similar point can be made about the homeostatic rule: why is 4Hz the target? Do things break down with a different target rate?*

The target of 4 Hz was used as it is in the range of *in vivo* recordings indicating that the spontaneous activity of pyramidal neurons ranges between 1–5 Hz:

- Burns, B. D., & Webb, A. C. (1976). The spontaneous activity of neurones in the cat’s cerebral cortex. *Proceedings of the Royal Society of London. Series B. Biological Sciences*, 194(1115), 211-223.
- Softky, W. R., & Koch, C. (1993). The highly irregular firing of cortical cells is inconsistent with temporal integration of random EPSPs. *Journal of neuroscience*, 13(1), 334-350.
- Wilson, F. A., O’scalaidhe, S. P., & Goldman-Rakic, P. S. (1994). Functional synergism between putative gamma-aminobutyrate-containing neurons and pyramidal neurons in prefrontal cortex. *Proceedings of the National Academy of Sciences*, 91(9), 4009-4013.
- Sakata, S., & Harris, K. D. (2012). Laminar-dependent effects of cortical state on auditory cortical spontaneous activity. *Frontiers in neural circuits*, 6, 109.
- Barth, A. L., & Poulet, J. F. (2012). Experimental evidence for sparse firing in the neocortex. *Trends in neurosciences*, 35(6), 345-355.

We did not test other target rates, but since the underlying neural mass model dynamics are known to undergo bifurcations for higher input currents (Wong KF, Wang XJ (2006) A recurrent network mechanism of time integration in perceptual decisions. *J Neurosci*.) it’s likely that also emerging dynamics are different when the dynamical regime of the model changes. This is generally true for any model parameter: since the model was described by Wong & Wang in 2006 it was used in several subsequent studies in that same regime that was configured to show biologically plausible behaviour. Outside of the biologically plausible regime “anything can happen” – but we didn’t specifically study this.

- *In general, how realistic are the simulated brain time series? While we are assured that the simulated rsFC matches that of the empirical time series (again, by design), real brain activity has many other characteristic behaviors, such as a particular structure in the autocorrelation / power spectrum. While it’s entirely out of scope to require the whole-brain model to produce realistic human brain time series, to what degree the model does so is important to acknowledge, given that the overarching question is “how to map human intelligence to machines”.*

Brain network models and the underlying neural mass or neural field models are specifically designed to simulate and explain the underlying biology. In contrast to, for example, artificial neural networks, biological neural models are specifically derived from biological spiking network experiments and simulations to mechanistically explain the activity of neural populations as we can observe them with various invasive and noninvasive methods – all based on and derived from our biophysical understanding of spiking neuronal structure, dynamics and functioning:

- Deco, Gustavo, et al. "The dynamic brain: from spiking neurons to neural masses and cortical fields." *PLoS computational biology* 4.8 (2008): e1000092.
- Breakspear, Michael. "Dynamic models of large-scale brain activity." *Nature neuroscience* 20.3 (2017): 340-352.

After the simulation of neural activity like synaptic input currents, synaptic ion channel activity or firing rates, the activity is input to so-called forward models like the Balloon Windkessel Model that describes the hemodynamic coupling between neural activity and blood flow:

- Friston, Karl J., et al. "Nonlinear responses in fMRI: the Balloon model, Volterra kernels, and other hemodynamics." *NeuroImage* 12.4 (2000): 466-477.

- Friston, Karl J., Lee Harrison, and Will Penny. "Dynamic causal modelling." *Neuroimage* 19.4 (2003): 1273-1302.

In short, spiking/mass/field models reproduce and predict typical neural activity features *per design* – they are specifically constructed to successively “reverse-engineer” biological brain structure and dynamics in the form of computational models.

While neural models generally focus on explaining generic patterns of ongoing dynamics, the actual moment-to-moment activity of a living brain is of course subject to many factors that are not readily available to the model (like the state of the entire rest of the body, cardiac and respiratory activity, etc.), but it may be relevant to mention that we specifically studied the quality of time series prediction in the same type of brain network model in another article, where we combined the model with empirical EEG data, and which allowed to characterize a mechanistic sequence of biological activity patterns from very fast (sub-millisecond firing rate patterns) to very slow (minutes to hours BOLD/functional connectivity dynamics):

- Schirner, M., McIntosh, A. R., Jirsa, V., Deco, G., & Ritter, P. (2018). Inferring multi-scale neural mechanisms with brain network modelling. *elife*, 7, e28927.

To specifically address the question by the Reviewer about the plausibility of time series, PSD and autocorrelation we added the new Supplementary Figure 10, which shows exemplary time series, as well as average PSD and autocorrelation plots of simulated versus empirical time series and we now added to the Discussion the following:

“Despite these limitations, BNMs are in contrast with artificial neural networks specifically designed to explain the underlying biology, using typically observed features of the empirical system as targets for validation and falsification (Supplementary Fig. 10) to achieve an incrementally improved computer model of the empirical system.”

- *there is very little detail or motivation provided for the reduced 2-node model: why PFC and PPC? Is there something special about the specific parameter values that make this model more akin to those two areas than, say, OFC and ACC? Or is it simply a generic two-node version of the full 379-node model?*

The two-node model is unrelated to the DM circuit and therefore unrelated to specific anatomical locations like PFC/PPC. Rather, the two-node model is a simplified version of the 379-node model to study the effect of E/I-balance in isolation and with a simple connection pattern. In contrast, the DM circuit is an established circuit for studying DM in a frontoparietal (PFC and PPC) circuit, motivated by empirical observation of prefrontal and parietal involvement in decision-making (please also see other comments on previous publications of the DM circuit above and referenced in the manuscript). We added new sentences to the description of the two-node model to make this clearer:

“To study how E/I-ratios modulate FC in isolation we tuned E/I-ratios from 0.01 to 100 in the two-node model. The two-node model is a simplified version of the 379-node large-scale brain model to study the effect of large-scale E/I-balance with a simpler network structure (Fig. 4c-j). The two-node model (equations 1-6) differed from the functional frontoparietal decision-making circuit⁹⁷ (DM circuit, equations 7-10) further introduced below. The two-node model simulated mutual and recurrent interaction between one excitatory and one inhibitory population as in the 379-nodes large-scale model, but with a simpler network of only two nodes to produce tuning curves (Fig. 4c-h). In contrast, the DM circuit is an existing frontoparietal circuit model to simulate winner-take-all competition resulting from cross-inhibition of two excitatory populations via one inhibitory population, which we studied in isolation (Fig. 5), and after coupling with the 379-nodes large-scale model to form the multiscale model (Fig. 6). Dynamics of the two-node model were identical to the full 379-regions model but with only two nodes i, j that had a mutual coupling strength of $C_{ij} = C_{ji} = 1$.”

- *critically, I'm actually not sure how the two-node (nor the subsequent “multiscale” 379-node model) is doing the task? Specifically, how are the “% correct decisions” and “integration time” values in Figure 3 derived? Apologies if I had missed this, but I could not find this in the method section nor in the main text, and I believe this is a central piece of information to evaluate the soundness of the authors’ claims and the mechanistic conclusions drawn from the model. A description of this is in words is found in the supplements (which is definitely not where it should be), but simply writing out the equations, and the inputs and outputs, would make this much easier to evaluate the soundness of the approach.*

We avoided extensive descriptions of the DM circuit (please see our other responses above) in the main text and rather put it in the Supplementary as the DM circuit is not a Result of our work, but we only re-used it from Murray et al. We now added the description for computing “% correct decisions” and “integration time” to the Methods section:

“Decision-making performance was computed as in the original publication of the DM circuit by Murray et al. by modelling the strength of evidence as an external current to the two parietal populations A_{PPC} and B_{PPC} as follows:

$$I_{app,i}^n = I_e \left(1 \pm \frac{c'}{100\%} \right), \quad (12)$$

where $I_e = 0.0118$ nA scales the overall strength of the input and $c' = 6.4\%$, referred to as the strength of evidence or contrast, determines which of the two populations A_{PPC} or B_{PPC} receives higher evidence (A_{PPC} received the higher evidence), which reflects the saliency of the target with respect to that of distractors. As in Murray et al.³⁷, when one of the two action populations A_{PFC} or B_{PFC} reaches a firing rate threshold of 40 Hz the decision for option A or B is taken and a reaction time is registered. We repeated the decision-making task 1000 times in order to compute the percentage of times for which the decision was made correctly (number of times A_{PFC} crossed the firing-rate threshold divided by the total number of trials) and the average time until the threshold was reached.”

Some other minor suggestions

- Figure 1. e-j showing the non-significant correlations (with fairly large magnitude of correlation) is a bit misleading since it's only clarified in the main text. It's easy to mistake a correlation value of 0.35 as notable, even if statistically indistinguishable from the null effect. I'd recommend removing these except for g.*

We now removed the non-significant correlations except for g from Fig. 1. We note that this comment helped us to identify and correct an error: we previously denoted Figure 1 panel j as subject-level correlations which were actually group-level correlations. To be consistent with the style from the top panel we now show group-level and subject-level results side by side in the updated Fig. 1.

- Could use more informative labels for legend in many places, instead of, e.g. PMAT24_A_CR, it'd be a lot easier for the reader to see % correct*

“% correct” and “PMAT24_A_CR” are two different metrics and cannot be identified by the same name. The latter is used to quantify human fluid intelligence following application of the PMAT24 test, while the former quantifies the performance of the decision-making model.

- Figure 3 could really use a circuit schematic, especially showing how the DM “task” is abstracted for the model*

We now provide the circuit schematic (former Supp. Fig. 2 d) in the main text as Fig. 2.

*Richard Gao, PhD
University of Tuebingen*

We would like to thank the Reviewer again for taking the time to provide detailed comments, which helped considerably to improve the manuscript.

REVIEWERS' COMMENTS

Reviewer #2 (Remarks to the Author):

The authors addressed all my comments. I have no further comments; thus, I endorse the current manuscript for publication.

Reviewer #3 (Remarks to the Author):

Thanks to the authors for responding to my comments, and for providing all the additional clarifications and detailed answers to my questions. I think the additions in the main text and figures are very helpful for contextualizing this study, at least for myself, but hopefully to other readers unfamiliar with the whole-brain modeling literature. Many of my questions/confusions were due to unfamiliarity with existing literature in this field, and many addressed by the additional references / context the authors provide in their response. I know it was probably frustrating, so I hope the authors don't take it personally (I don't want to be "that reviewer" either).

In particular:

- the clarification that the 2-node model and the DM circuit are different and unrelated models was very helpful. I was clearly confused about this, sorry
- the discussion of limitations of the BNM approach is very helpful, and so are the references pointing to the "standards", so to say, in DCM and other types of work. Maybe it's obvious to the authors, but I think it's helpful for a broad audience.
- the schematic in Figure 2 is quite helpful.
- the motivation for why FC is used is clearer in both the authors' response and in the main text
- tuning EI ratio or EI balance, in many other branches of neuroscience literature, refers to tuning inhibition strength directly. The additional references on feedforward inhibition, and the definition as the ratio, is helpful for clarifying that this is not the case.

A few disagreements:

- on the comment that "the identified mechanistic link exists independent of any correlations with empirical data": yes, of course the model is self-consistent without considering the data, but we're not interested in the model itself, but what one learns from the model about the brain. Obviously we care about the actual correlation between FC and behavior in the real data, that's what motivates the model

in the first place. Had there been no correlation, it wouldn't have mattered that the model produces a correlation between simulated FC and task performance, that's all I meant. In any case, the additional references arguing that $r=0.15$ is actually quite large provides sufficient context.

- on the comment "we never claimed that the model exhibits fluid intelligence". Line 26-28 in the original abstracts states "Model simulations revealed a mechanistic link between EI balance, fluid intelligence, decision-making...". I took this to mean that the model exhibits fluid intelligence and is mechanistically linked to EI balance (in the model), because, as per the authors' response to the previous comment "The identified mechanistic link exists [...] it describes a chain of events that is explicitly implemented as a computational model that can be reproduced and demonstrated, capturing the entire logic of the theory."

- yes, "that other mechanisms exist doesn't automatically invalidate the particular mechanism we are proposing", but the claims made in the paper is clearly beyond the level of "this is a potential mechanism", and therefore some level of sensitivity analysis to static parameters (like tuned rate of 4Hz) or other mechanisms, and discussion thereof, seems appropriate?

Anyway, these are just small nitpicks, no need for another round of response. Good luck to the authors and congratulations again on the interesting work!

Richard Gao

Point-by-point response to reviewer's comments

We would like to thank the Reviewers for providing a second round of reviews and for their positive evaluation of the manuscript.
Thank you for your time and effort!

REVIEWERS' COMMENTS

Reviewer #2 (Remarks to the Author):

The authors addressed all my comments. I have no further comments; thus, I endorse the current manuscript for publication.

We would like to thank Reviewer #2 for their positive evaluation and endorsement. Thank you again!

Reviewer #3 (Remarks to the Author):

Thanks to the authors for responding to my comments, and for providing all the additional clarifications and detailed answers to my questions. I think the additions in the main text and figures are very helpful for contextualizing this study, at least for myself, but hopefully to other readers unfamiliar with the whole-brain modeling literature. Many of my questions/confusions were due to unfamiliarity with existing literature in this field, and many addressed by the additional references / context the authors provide in their response. I know it was probably frustrating, so I hope the authors don't take it personally (I don't want to be "that reviewer" either).

In particular:

- the clarification that the 2-node model and the DM circuit are different and unrelated models was very helpful. I was clearly confused about this, sorry**
- the discussion of limitations of the BNM approach is very helpful, and so are the references pointing to the "standards", so to say, in DCM and other types of work. Maybe it's obvious to the authors, but I think it's helpful for a broad audience.**
- the schematic in Figure 2 is quite helpful.**
- the motivation for why FC is used is clearer in both the authors' response and in the main text**
- tuning EI ratio or EI balance, in many other branches of neuroscience literature, refers to tuning inhibition strength directly. The additional references on feedforward inhibition, and the definition as the ratio, is helpful for clarifying that this is not the case.**

A few disagreements:

- on the comment that “the identified mechanistic link exists independent of any correlations with empirical data”: yes, of course the model is self-consistent without considering the data, but we’re not interested in the model itself, but what one learns from the model about the brain. Obviously we care about the actual correlation between FC and behavior in the real data, that’s what motivates the model in the first place. Had there been no correlation, it wouldn’t have mattered that the model produces a correlation between simulated FC and task performance, that’s all I meant. In any case, the additional references arguing that $r=0.15$ is actually quite large provides sufficient context.

- on the comment “we never claimed that the model exhibits fluid intelligence”. Line 26-28 in the original abstracts states “Model simulations revealed a mechanistic link between EI balance, fluid intelligence, decision-making...”. I took this to mean that the model exhibits fluid intelligence and is mechanistically linked to EI balance (in the model), because, as per the authors’ response to the previous comment “The identified mechanistic link exists [...] it describes a chain of events that is explicitly implemented as a computational model that can be reproduced and demonstrated, capturing the entire logic of the theory.”

- yes, “that other mechanisms exist doesn’t automatically invalidate the particular mechanism we are proposing”, but the claims made in the paper is clearly beyond the level of “this is a potential mechanism”, and therefore some level of sensitivity analysis to static parameters (like tuned rate of 4Hz) or other mechanisms, and discussion thereof, seems appropriate?

Anyway, these are just small nitpicks, no need for another round of response. Good luck to the authors and congratulations again on the interesting work!

Richard Gao

We would like to thank Reviewer #3 for their positive evaluation and their kind endorsement. We appreciated the direct style of the review and welcomed the honest reflection of concerns as we found it helpful to identify problems with the presentation. Thank you again!